



**Monte-Carlo Analysis of Asymmetry in Three-Site Relaxation Exchange:**
**Probing Detailed Balance**
Bernhard Blümich,[1] Matthew Parziale,[2] and Matthew Augustine[2]
[1] Institut für Technische und Makromolekulare Chemie, RWTH Aachen University,
Worringer Weg 2, 52074 Aachen, Germany
[2] Department of Chemistry, UC Davis, One Shields Avenue, Davis, CA 95616, USA
Corresponding author: Bernhard Blümich, bluemich@itmc.rwth-aachen.de
The question is investigated if three-site diffusive relaxation exchange in
thermodynamic equilibrium can lead to exchange maps, which are asymmetric for
fluids confined to pores. Asymmetry reports circular flow of particles between the
relaxation sites which disagrees with detailed balance according to which the particle
exchange between any pair of sites must be balanced. Vacancy diffusion and gas
diffusion of particles confined to two-dimensional pores were modeled in Monte-Carlo
simulations. For each particle move in vacancy diffusion on a 2D checkerboard grid,
one of the eight neighboring destination cells was identified on the basis the jump
probability calculated from an empirical approximation of the free energy. Gas diffusion
was simulated without thermodynamic interaction on a simulation grid which was up to
$10^4$ times finer than the particle diameter. It was found that up to 1% of all particles
moves coherently in closed paths. This motion is attributed to pore resonance
corresponding to diffusion eigenmodes. The study shows that detailed balance of
multi-site exchange does not apply for a small fraction of particles when the exchange
is impacted by topological constraints.

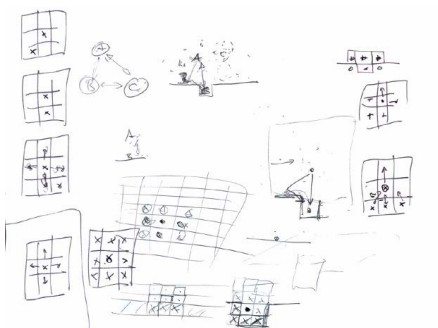


Graphical abstract: Draft of particle exchange on a 2D checkerboard grid





## 1. Introduction

Exchange phenomena are an essential ingredient of diffusion and spreading phenomena, which are abundant in nature and govern the evolution of tangible and intangible objects and goods (Bunde et al., 2018). Nuclear Magnetic Resonance provides particularly powerful methodologies to investigate molecular exchange processes (Ernst et al., 1987; Callaghan, 2011). Slow molecular exchange on the millisecond time scale is studied by two-dimensional exchange NMR, i. e. by chemical exchange spectroscopy for rotational motion (Jeener et al., 1979) and by exchange relaxometry for translational motion (Lee et al., 1993). In equilibrium the nature of the exchange processes is commonly understood to be random Brownian motion, and the 2D exchange maps are expected to be symmetric with respect to their diagonal. On the other hand, exchange in non-equilibrium leads to asymmetry in 2D NMR maps. This has been observed, for example, in 2D chemical exchange spectra for chemical reactions involving different sites (Lacabanne et al., 2022), for the spread of hyperpolarization by spin diffusion (Björgvinsdóttir et al., 2021), and for slow flow across porous media in relaxation exchange maps (Olaru et al., 2012).

The kinetics of transitions or exchange between discrete states driven by random processes are described by (van Kampen, 1992)

$$\frac{\mathrm{d}M_i(t)}{\mathrm{d}t} = \sum_j \{k_{ij}M_j(t) - k_{ji}M_i(t)\}, \tag{1}$$

where $M_i$ are populations or longitudinal magnetization components collected in the vector $\boldsymbol{M}$, and $k_{ij}$ are the exchange rates equivalent to the transition probabilities from state $j$ to state $i$, which are collected in the kinetic exchange matrix $\mathbf{k}$. In equilibrium

$$\frac{\mathrm{d}M_i(t)}{\mathrm{d}t} = 0, \tag{2}$$

and the number of all particles arriving at site $i$ from sites $j$ is equal to the number of all particles leaving from site $i$ to sites $j$ so that the total mass is conserved.

As a result of mass balance, two-site exchange between states or sites A and B always leads to symmetric 2D NMR exchange maps in thermodynamic equilibrium as the number $k_{\mathrm{BA}}M_\mathrm{A}$ of particles populating site B by leaving site A per unit time is equal to the number of particles $k_{\mathrm{AB}}M_\mathrm{B}$ leaving site B and populating site A per unit of time. This number is the product of the rate $k_{\mathrm{BA}}$ for transitions from site A to site B times the population $M_\mathrm{A}$ of site A. The relationship $k_{\mathrm{BA}}M_\mathrm{A} = k_{\mathrm{AB}}M_\mathrm{B}$ is known as the 'principle of detailed balance'. It is usually taken to also apply to rate processes involving more than two sites.



By example of mass-balanced equilibrium diffusion between three sites
(Sanstrom, 1983), eqn. (1) becomes
$k_{21}M_1 + k_{31}M_1 = k_{12}M_2 + k_{13}M_3,$
$k_{12}M_2 + k_{32}M_2 = k_{21}M_1 + k_{23}M_3,$                (3)
$k_{13}M_3 + k_{23}M_3 = k_{31}M_1 + k_{32}M_2,$
or equivalently, mass balance requires
$k_{12}M_2 - k_{21}M_1 = k_{31}M_1 - k_{13}M_3 = k_{23}M_3 - k_{32}M_2.$                (4)
Normalization of this expression to the total number of exchanges per unit time defines
the asymmetry parameter $a_{sy}$ to be used below,
$(k_{23}M_3 - k_{32}M_2)/[(1 \quad 1 \quad 1)\,\mathbf{k}\,\mathbf{M}] \stackrel{\text{def}}{=} a_{sy}.$                (5)
While mass balance (4) is a necessary condition for dynamic equilibrium, detailed
balance, on the other hand, is a stronger condition and requires
$a_{sy} = 0.$                (6)
Detailed balance had been introduced by Maxwell in 1867 based on 'sufficient
reason' in his derivation of the speed distribution of gas atoms considering the speed
exchange between colliding gas atoms in thermodynamic equilibrium (Maxwell, 1867).
An intriguing consequence of the exchange being balanced in detail between particles
A and B amounts to the impossibility of assigning positive time to either velocity
exchange from A to B or B to A on the particle scale of the exchange process, thus
admitting negative time or time reversal. In 1872 Boltzmann showed in an elaborate
treatment, that Maxwell's speed distribution also applies to polyatomic gas molecules
(Boltzmann, 1872). Furthermore, in 1917 Einstein derived Planck's law of black-body
radiation as a balanced energy exchange between quantized radiation and matter
underlining the striking similarity to Maxwell's speed distribution of gas atoms (Einstein,
1917). He concludes "Indem Energie und Impuls aufs engste miteinander verknüpft
sind, kann deshalb eine Theorie erst dann als berechtigt angesehen werden, wenn
gezeigt ist, daß die nach ihr von der Strahlung auf die Materie übertragenen Impulse
zu solchen Bewegungen führen, wie sie die Wärmetheorie verlangt," (Since energy
and momentum are intimately connected, a theory can only then be considered
justified, when it has been shown, that according to it the momenta of the radiation
transferred to the matter lead to such motions as demanded by the theory of heat.)
In his work extending Maxwell's speed distribution to polyatomic gas molecules
Boltzmann considered molecules in a container whereby the walls reflect the
molecules like elastic balls: "Bezüglich der Gefäßwände, welche das Gas





umschließen, will ich jedoch voraussetzen, dass die Moleküle an denselben wie
elastische Kugeln reflektiert werden. … Die Wände stören nicht, da an ihnen die
Moleküle wie elastische Kugeln reflektiert werden; also geradeso von ihnen
zurücktreten, als ob der Raum jenseits der Wände von gleich beschaffenem Gase
erfüllt wäre." (Concerning the container walls which enclose the gas, I want to presume
that the molecules are reflected from them like elastic balls. …. The walls do not
interfere, because the molecules are reflected from them like elastic balls; that is,
recede from them just like that, as if the space beyond the walls would be filled with
similarly conditioned gas.) Moreover, the interaction between gas molecules can be of
any type. While he claimed that any other interaction between walls and molecules
would lead to the same result albeit at loss of simplicity, the perfectly elastic reflections
of the gas molecules at the walls would eliminate the topological constraints of the box
on their motion. Boltzmann obtained the same speed distribution for polyatomic
molecules with internal degrees of freedom as Maxwell had for atoms based on
detailed balance of speed exchange. In the simulations reported below, the motion of
molecules is considered for which the interactions with the walls are the same as those
among the molecules. Understanding confined diffusion (Valiullin, 2017) is important
from a general point of view because the motion of molecules without topological
constraints is an ideal limit which cannot perfectly be realized in practice although it
may be realized within experimental uncertainty.

Two-site exchange processes will always be symmetric in thermodynamic

equilibrium. This situation has been evaluated analytically for NMR relaxation
exchange of fluids in porous media (McDonald, 2005). Yet multi-site relaxation-
exchange NMR maps (Van Landeghem, 2010) can from a mathematical point of view
be asymmetric. For example, the transverse magnetization $s(t_1, t_2)$ from a three-site
$T_2$-$T_2$ relaxation exchange NMR experiment (Gao and Blümich, 2020),

$$s(t_1, t_2) = (1,1,1)e^{-(\mathbf{R}_2+\mathbf{k})t_2}e^{-(\mathbf{R}_1+\mathbf{k})t_m}e^{-(\mathbf{R}_2+\mathbf{k})t_1}\boldsymbol{M}(t_0), \tag{7}$$

has been simulated to model an experimentally observed asymmetric three-site $T_2$-$T_2$
NMR exchange map of water molecules saturating $Al_2O_3$ powder with the three
relaxation sites corresponding to bulk water, water molecules on the surface of the
powder particles and water molecules inside the surface pores (Fig. 1). Here $\boldsymbol{M}(t_0)$ is
the initial vector of transverse magnetization components from relaxation sites 1, 2 and
3 generated from longitudinal thermodynamic equilibrium magnetization with a 90°
pulse at the beginning of the experiment at time $t_0$, and $t_1, t_m, t_2$ are the evolution,



mixing, and detection time intervals of the 2D NMR experiment, respectively
(Callaghan, 2011; Lee et al., 1993). Apart from the relaxation-rate matrices $\mathbf{R}_1$ and $\mathbf{R}_2$,
and the kinetic matrix $\mathbf{k}$, the best match obtained by forward simulation returned the
peak integrals revealing an asymmetry parameter of $a_{\mathrm{sy}} = -1.2\%$. This asymmetry of
the forward and backward particle jumps between two sites specifies the percentage
of circular exchange events per unit time between the three sites in thermodynamic
equilibrium (Fig. 1).

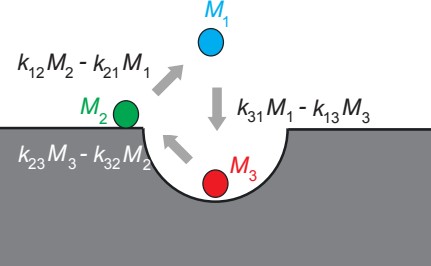


Figure 1. Asymmetry in three-site diffusion-mediated exchange indicates coherent
circular motion by example of water molecules in contact with porous $Al_2O_3$ grains.
Three water populations $M_j$ are identified by different NMR relaxation times and color.
They are molecules in the bulk (1), molecules on the particle surface (2) and molecules
in the pores (3). The exchange rate constants are $k_{ji}$. The net particle flux between two
sites differs from zero. The net mass of all molecules participating in the exchange is
conserved. The figure illustrates positive $a_{\mathrm{sy}}$.

While it can be argued that the experimental value of asymmetry lies within the

measurement uncertainty, molecular dynamics simulations are reported and
discussed below to investigate whether asymmetry in three-site exchange is just a
mathematical peculiarity or a realistic physical phenomenon. This study is thought
provoking in view of the fact, that asymmetric three-site exchange disagrees with
detailed balance of the exchange between any pair of sites and needs to be explained
by circular diffusion on the pore scale in thermodynamic equilibrium. Such motion
resembles that of a rachet which Feynman has argued to disagree with the second law
of thermodynamics (Feynman et al., 1966).

**2. Modelling confined diffusion**
**2.1 Vacancy diffusion: Random particle jumps on a 2D checkerboard**
Random jumps of particles from occupied sites to vacant sites were simulated with a
Monte-Carlo algorithm (Grebenkov, 2011; Hughes, 1995; Sabelfeld, 1991) in a





confined space on a checkerboard. The algorithm models vacancy diffusion (Seitz,
1948) encountered in metals and alloys but the particles perform the jumps rather than
the vacancies. To keep the simulation simple, it is limited to jumps on a 2D 3×3 Moore
lattice of range 1 (Wolf-Gladrow, 2000) following rules of the game of life (Wolf-
Gladrow, 2000; Bialnicki-Birula, 2004). Here the center particle can jump to any of its
8 neighbors. Different neighborhoods of range 1 were tested (Fig. S1) (Bialnicki-Birula,
2004), but only the Moore neighborhood having the highest symmetry of all
neighborhoods, produced data consistent with Eqn. (4). Topological constraints are
introduced which set boundaries to the jump space. Initially, the available cells inside
the jump space on the grid are populated randomly with particles up to a specified
particle density. Particles in the bulk are indexed 1, and two distinct boundary sections
are indexed 2 and 3, giving three environments between which randomly selected
particles can move. A particle jumping from environment $j$ to $i$ is counted by
incrementing the element $ij$ of a 3×3 jump matrix by 1. If the particle environment does
not change with the jump, the respective diagonal element is incremented.

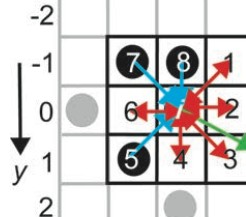

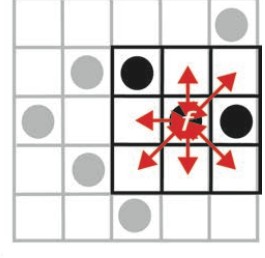

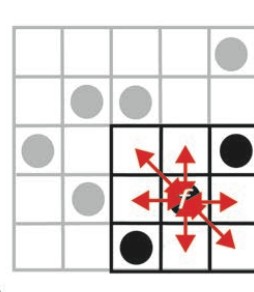

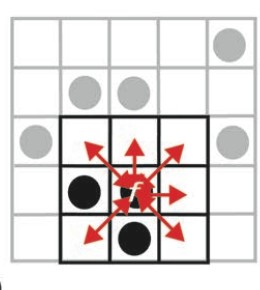

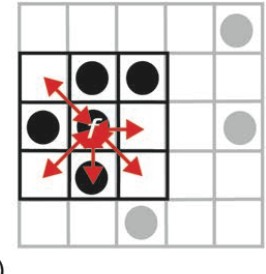


Figure 2. Checkerboard randomly occupied by particles represented by filled circles.
a) The cells surrounding the initial particle position $i$ are numbered clockwise from 1 to
8. Cells 5,7, and 8 are occupied, the others are empty. The force (green arrow) on the
center particle is calculated as the sum of forces exerted from all particles in the
occupied nearest neighbor cells (blue arrows). The entropy is estimated from the sum





of distances to all neighboring free cells (red double arrows). b-f) The center particle in
a) can jump to any of the free cells 1, 2, 3, 4, and 6, each of which has its own entropy.
The final position *f* of the jump is identified by evaluating the jump probability based on
a simple heuristic model of the free jump-energy difference.

A particle on a square grid has eight nearest neighbor cells to which it can jump.
(Fig. 2a). If more than one out of all neighbor cells are empty and thus available as
destination cells for the jump, the destination cell is identified by evaluating the jump
probability for each destination cell based on a simple heuristic model of the free jump-
energy difference (Fig. 2b-f). At constant temperature and constant volume, the particle
motion is governed by the Helmholtz free energy $A = U - T\,S$, where $U$ is internal
energy, $T$ is temperature, and $S$ is entropy. In thermodynamic equilibrium, $A$ is at its
minimum. The probability of a particle moving from one cell to another is given by the
Boltzmann distribution $p = \exp\left\{-\frac{\Delta A}{k_B T}\right\}$, where $\Delta A = \Delta U - T\,\Delta S$.
The free energy change $\Delta A = \Delta U - T\,\Delta S$ is determined from crude models of the
internal energy change $\Delta U = \boldsymbol{F}\,\Delta \boldsymbol{R}$ defined by the repulsive net force $\boldsymbol{F}$ exerted from
all neighboring particles on the particle at stake and the length $|\Delta\boldsymbol{R}|$ of the jump to the
next cell, the temperature $T$, and the entropy change $\Delta S$. The force $\boldsymbol{F}$ between two
particles follows Newton's inverse square distance law. It is proportional to $\frac{1}{|\Delta\boldsymbol{R}|^2}$ in the
direction of $\frac{1}{|\Delta\boldsymbol{R}|}\Delta\boldsymbol{R}$ from an occupied cell *j* to the particle *i* under consideration. The total
force the particle *i* experiences is the vector sum of the forces exerted from the particles
*j* in all occupied neighbor cells (Fig. 2a),
$$\boldsymbol{F}_i = \sum_j \frac{1}{\left|\Delta\boldsymbol{s}_{j,i}\right|^3}\Delta\boldsymbol{R}_{j,i}. \tag{8}$$
The internal energy change $\Delta U_{f,i}$ is computed for each potential jump from the initial
occupied cell *i* to the final empty cell *f* by the product of the net force $\boldsymbol{F}_i$ with the vector
$\Delta\boldsymbol{R}_{f,i}$ connecting the centers of the initial cell *i* and the final cell *f*.
The entropy change $\Delta S = S_f - S_i$ is the difference between the entropies of the
particle with its eight nearest neighbors for the final state *f* and the initial state *i*. It is
estimated by the sum of the step lengths $R_{f,i} = \left|\Delta\boldsymbol{R}_{f,i}\right|$ of the particle *i* to its unoccupied
next nearest neighbor cells *f*,
$$S_i = \sum_f \Delta R_{f,i}. \tag{9}$$
In case a neighbor cell is occupied, $\Delta\boldsymbol{R}_{f,i} = 0$. Detailed examples are worked out in the
supplementary information (Fig. S2).



With these definitions jump probabilities $p = \exp\left\{-\frac{\Delta A}{k_B T}\right\}$ can be calculated. In
each jump step, an initially occupied cell *i* is selected at random and $p$ is evaluated for
all possible jumps to neighboring empty cells as potential final cells *f*. If for one or more
jumps $p \geq 1$, the destination cell of the jump is picked at random from this subset of all
potential jumps. If all neighbor cells are occupied, $p = 0$, and no jump is possible. If
$0 < p < 1$ the destination cell is chosen at random from all those with the same largest
jump probability $p < 1$.
In the simulations reported below, the Boltzmann constant $k_B$ has been set to 1
and so has the shortest distance between neighboring cells. Depending on the type of
cell, periodic boundary conditions (Fig. 3a) or rigid boundaries (Fig. 3b) were
employed. A cell boundary has been treated just like an occupied cell with the same
definition of the free energy in the calculation of the jump probability.
The simulations were carried out with a program written in Matlab R2020a by The
MathWorks Inc. on an Apple MacBook Pro 2.4 GHz having an Intel Quad-Core i5
processor. Typically, $10^7$ jumps were simulated in one run taking 75 seconds.

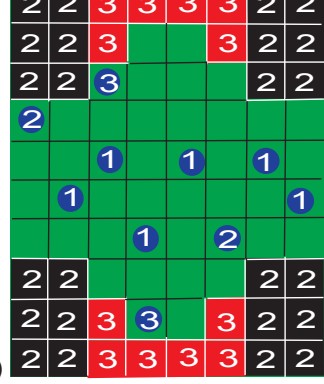  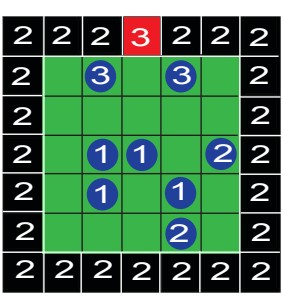

a)                                                b)

Figure 3. Examples of pore models for two-dimensional three-site exchange based on
a checkerboard grid. Particles can occupy one cell and jump to a neighboring empty
one based on the free-energy difference of the jump. a) Porous solid. The boundaries
right and left are periodic. The boundaries top and bottom are rigid. Depending on their
next neighbors in the first coordination shell, the particle-relaxation environments are
identified as bulk (1), surface (2) or pore (3). b) Small square pore with an active site.
The bulk (1), the walls (2), and the active site (3) have different relaxation properties.
If a particle cell contacts two different relaxation sites, the higher number overrides the
lower number when identifying its relaxation environment.

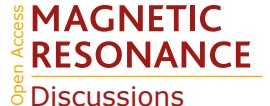

**2.2 Gas diffusion**
The gas diffusion calculations explore similar pore size and occupancy. Here the
motion of circular particles with diameter equal to the cell size was accomplished by
propagating an initial Maxwell–Boltzmann distribution of particle speeds for random
initial positions in a Monte Carlo fashion based on instantaneous collisional forces. The
collisions change both the direction and velocity of the particles at each of the $10^9$
constant time increments used here. These elastic collisions with other particles and
the wall are mediated by the particle size, which is set to be a fraction of the pore side
length of one. This means that a square pore with a five-particle diameter side length
is populated with particles having a diameter of 1/5. To compare the continuous
positional output of this model to vacancy diffusion, a two-dimensional square grid with
cell size set by the particle diameter is imposed on the entire pore. The quasi-
continuous positional output is then binned into these cells and compared to the binned
positions from the previous observation to determine if particles translated between
the main pore volume, pore wall, and active site. The translational information is used
to assign estimates of the jump-matrix elements and thus the asymmetry parameter
$a_{sy}$. The distribution of particles in the pore is recorded at constant time intervals,
whereas for vacancy diffusion it is recorded after each jump.

**3. Results**
Two different pore geometries were analyzed. Initially the simulation was executed for
a pore geometry (Fig. 3a) which approximates the pore structure of Fig. 1 and which
is hypothesized to explain the observed asymmetry of water diffusing in a porous $Al_2O_3$
grain pack (Gao and Blümich, 2020). It is mirrored vertically to double the probability
of particles entering the dent (relaxation site 3) in the otherwise straight surface
(relaxation site 2). The bulk of the particles defines relaxation site 1. This complex pore
structure was studied first, and the simulations revealed the existence of asymmetric
exchange. To understand the essence of the asymmetry a simple square pore with an
active site was studied in detail modelling confined diffusion in a small pore. The bulk,
the walls and the active site have different relaxation properties (Fig. 3b). For both
structures, the asymmetry parameter $a_{sy}$ was evaluated for vacancy diffusion as a
function of temperature and pressure. The results for the complex pore a reported in
the supplementary material, whereas those for the simple square pore are reported in
the main text here. Pressure was varied in terms of the population density measured



as the fraction of cells occupied in the pore. At certain temperatures and pressures
also the autocorrelation function of the occupation-time track of a particular cell and its
Fourier transform were determined. Striking features observed in vacancy diffusion
were subsequently modelled for gas diffusion in the square pore.

Relevant results for the square pore (Fig. 3b) are summarized in six graphs in
Fig. 4. The asymmetry parameter varies strongly with temperature $T$ (Figs. 4a,b) and
pressure corresponding to population density $P$ (Figs. 4e,f). All parameters are relative
quantities without units. The top three graphs a), b) and c) show the variation of $a_{sy}$
with temperature for a population fraction of 0.3 corresponding that of a gas. The
asymmetry parameter assumes only negative values in an abrupt but reproducible
manner in the range of -0.8% < $a_{sy}$ < 0.0% for repulsive interaction (Fig. 4a), i. e. for
the definition of the force between particles as illustrated in Fig. 2a. With reference to
Fig. 1, negative $a_{sy}$ reports that the straight exit route from the active site towards the
center of the pore is preferred over the detour via the pore wall. When the interaction
is changed from repulsive to attractive by inverting the sign of $\Delta U$ in the expression for
the free energy, the asymmetry parameter varies as well, however, only between -
0.3% < $a_{sy}$ < 0.0% (Fig. 4b). In either case, the asymmetry parameter varies with the
thermodynamic conditions. It is concluded, that for this small pore, up to about 1% of
all jumps on the checkerboard can proceed in an ordered circular fashion between the
three sites. Similar behavior is observed for the complex pore of Fig. 3a as illustrated
in Fig. S3 in the supplement.

At the extrema of the $a_{sy}(T)$ curves in Figs. 4a,b the dependence of the
asymmetry parameters on population density was investigated (Figs. 4d-f). The
variations with population density are smoother than those with temperature.
Significant negative asymmetry results at intermediate pressure, while at low and high
pressure, the asymmetry is small (Fig. 4d,e). At higher temperature and high pressure,
small positive $a_{sy}$ is observed (Fig. 4d, $T$ = 0.8, $P$ = 0.8). If the interaction between
particles and walls is turned off, i. e. $\Delta A = 0$, essentially noise more than two orders of
magnitude smaller than with particle interaction is observed for the exchange
asymmetry determined from $10^6$ jumps when varying $T$ and $P$, however, with a small
bias towards negative $a_{sy}$ (Figs. 4c,f). This suggests that the non-zero values for $a_{sy}$
reported in Figs. 4a,b,d,e are trustworthy.




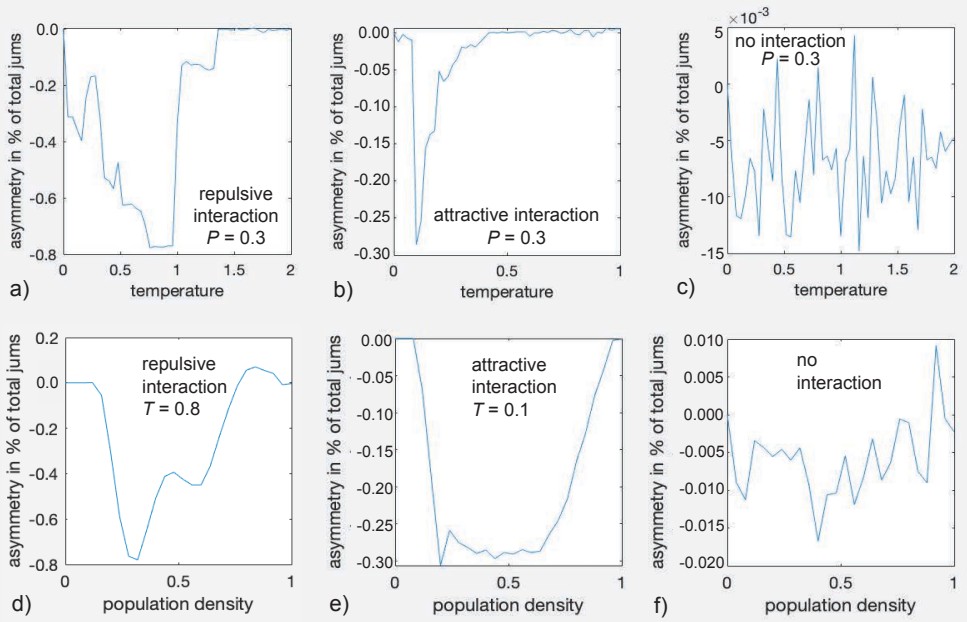

Figure 4. Asymmetry parameters $a_{sy}$ for diffusion inside the small rectangular pore depicted in Fig. 3b as a function of temperature $T$ (a-c) and pressure $P$ (d-f). a) $a_{sy}(T)$ for repulsive interaction at $P = 0.3$. b) $a_{sy}(T)$ for attractive interaction at $P = 0.3$. c) $a_{sy}(T)$ without interaction. d) $a_{sy}(P)$ for attractive interaction at $T = 0.8$. e) $a_{sy}(P)$ for attractive interaction at $T = 0.1$. f) $a_{sy}(P)$ without interaction.

To shed further light on the origin of the asymmetry, autocorrelation functions of the occupation-time tracks of selected cells in the pore were computed and Fourier transformed (Fig. 5). The occupation-time track was calibrated to zero mean for purely random occupation, i. e. it contained the negative population density when it was empty and the complement of the population density to one when the cell was occupied. The faster the autocorrelation function decays, the less coherent the cell population fluctuates and the broader is its Fourier transform, i. e. the transfer function (Fig. 5b,c). A constant offset of the autocorrelation function shows that the time-average population in the cell differs from the mean population of the pore (Fig. 5a,b). This offset produces a spike at zero frequency in the transfer functions. Subtracting the offsets from the autocorrelation functions and scaling the resulting functions to the same amplitude reveals different decays in different cells and thus variations in particle dynamics across the pore (inset in Fig. 5c, middle). These dynamics cannot readily be measured for a single cell in the pore, although an average over all cells and pores in the measurement volume would be amenable to experiment by probing the particle





dynamics with CPMG measurements in magnetic field gradients at variable echo time.
Such measurements provide the frequency-dependent diffusion coefficient as the
Fourier transform of the velocity autocorrelation function (Stepišnik et al., 2014,
Callaghan and Stepišnik, 1995; Parsons et al., 2006).

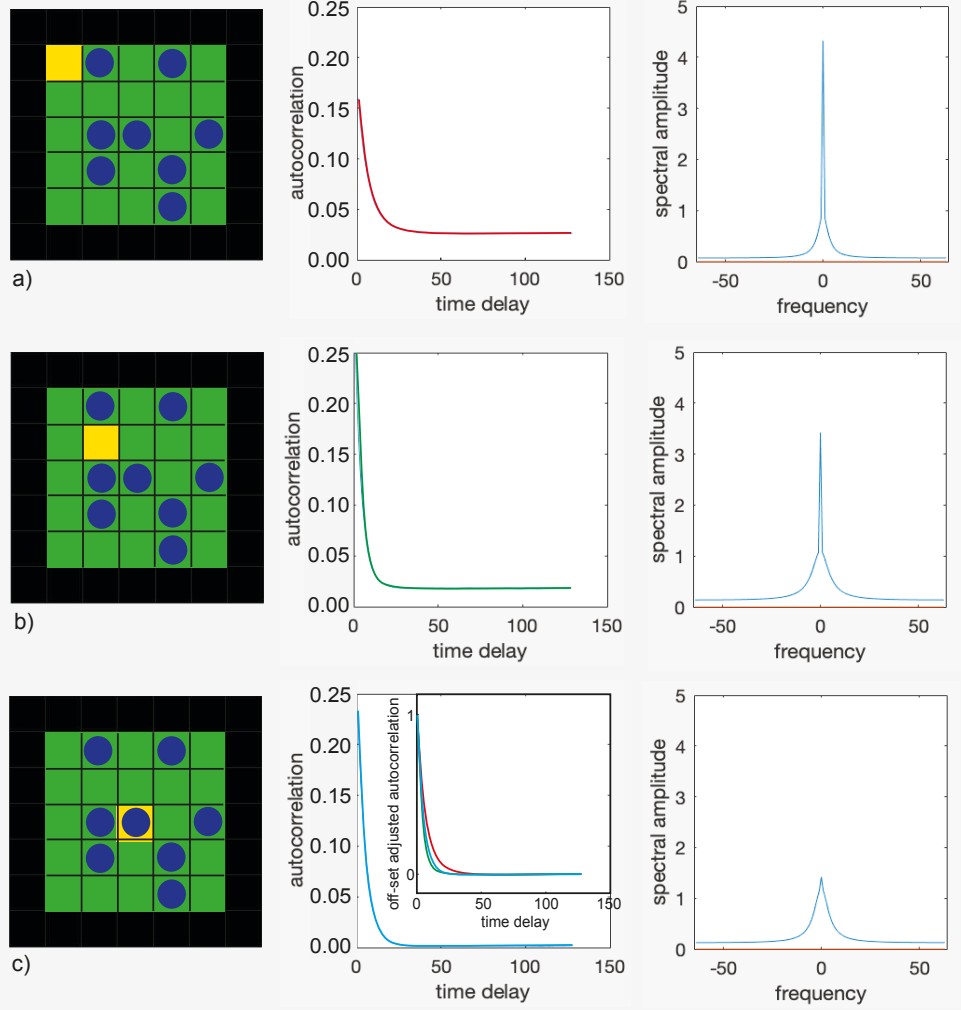


Figure 5. Autocorrelation functions (center) of the occupancy of the yellow cells (left)
and the real parts of their Fourier transforms (right) for repulsive interaction at $T = 0.1$
and $P = 0.3$. a) Corner cell. b) Off-center cell. c) Center cell. The inset in the middle
compares the decays of all three autocorrelation functions after subtraction of the
offsets.

While the autocorrelation function is difficult to probe experimentally, the

asymmetry parameter $a_{sy}$, on the other hand, also probes the particle dynamics and



could be investigated experimentally in a straight-forward manner by relaxation-
exchange NMR experiments provided the signal-to-noise ratio is good enough. The
parameter depends on the location of the relaxation center in the pore wall (Fig. 6).
This dependence has been verified to be identical for all walls of the square pore.
Moreover, it exhibits mirror symmetry about the center position (Fig. 6e). For vacancy
diffusion in a $5 \times 5$ square pore with walls 7 cells wide (Fig. 6a,b), $a_{\mathrm{sy}}$ varies consistently
with position irrespective of the particle interaction being positive, negative or zero but
differs strongly in magnitude. It is highest at the corner positions and lowest at the
center position. For zero particle interaction, $a_{\mathrm{sy}}$ is more than an order of magnitude
smaller than for repulsive interaction, so that the number of particle jumps had to be
increased to $10^9$ resulting in 3 h computation time for each data point in the
corresponding trace (black) Fig. 6e. Interestingly, $a_{\mathrm{sy}}$ for gas diffusion without particle
interaction (green, Fig. 6e) varies in a fashion similar to that for vacancy diffusion, is of
magnitude comparable to that of vacancy diffusion (black, Fig, 6e), but does not
change sign with position of the active site in the pore wall. In all cases the precision
of the asymmetry parameter $a_{\mathrm{sy}}$ obtained in the simulations exceeds the second
relevant digit.

The particle dynamics manifested in $a_{\mathrm{sy}}$ are accompanied by variations of the

average population density across the pore which is depleted in the first particle layer
at the pore wall, enhanced in the next layer, and tapers off towards the pore center in
both cases (Figs. 6b,d, Fig. S4). The densities vary in a similar fashion across the pore
for both types of diffusion albeit having somewhat different values as can be verified
by close inspection of the numbers in each cell in Figs. 6b,d.

The maps in Fig. 6b,d revealing the deviation of local population density from

average population density were calculated by summing the 2D maps of particle
distributions after each jump, normalizing the resultant maps to the number of jumps
and the particle density and subtracting the average mean expected for a constant
particle density across all cells in the pore. Further maps of population density
variations for the two different pores of Fig. 3 with other sizes and interaction
parameters are summarized in Fig. S4 of the supplement. While the particle density
varies less with temperature for vacancy diffusion, different density patterns are found
at different pressures. The strongest density variations are near the pore wall whether
the interaction is repulsive, zero, or attractive, which becomes particularly evident for
larger pores (Figs. S4b,d,e). At low density and in the absence of particle interactions





the main features of the density maps are strikingly similar for vacancy diffusion (Figs.
S4b) and gas diffusion (Figs. S4d). The particle density is strongly depleted at the pore
corners and near the wall and significantly increased in the next particle layer (Figs.
S4e,f). This behavior can be expected in vacancy diffusion when considering, that a
jump to an empty pore away from the wall can happen from all directions, while an
empty pore at the wall cannot be populated from the side of the wall. Also, for gas
diffusion in continuous space, a spherical particle cannot get to the wall closer than
half its diameter. For interacting particles, this concentration variation is carried forward
in vacancy diffusion with increasing distance from the wall leading to concentration
waves which taper off towards the center of the pore and interfere with each other
coming from different directions. For small pores interference patterns dominate the
density distribution across the pore (Figs. 6b,d and Figs. S4a,c). For noninteracting
particles, the decay of the concentration wave towards the pore center is fast with few
to no oscillations towards the pore center, while the oscillations are enhanced by
thermodynamic interactions between the particles (Fig. S4d, $P = 0.2$). In particular, the
population density at the active site in the dent of the complex pore of Fig. 3a depends
on the thermodynamic parameters $P$ and $T$ (Figs. S4a,b). It needs to be investigated
further how much the NMR relaxation times vary with the position in the standing
concentration wave and the associated particle dynamics from the pore wall to the
center (Bytchenkoff and Rodts, 2011).

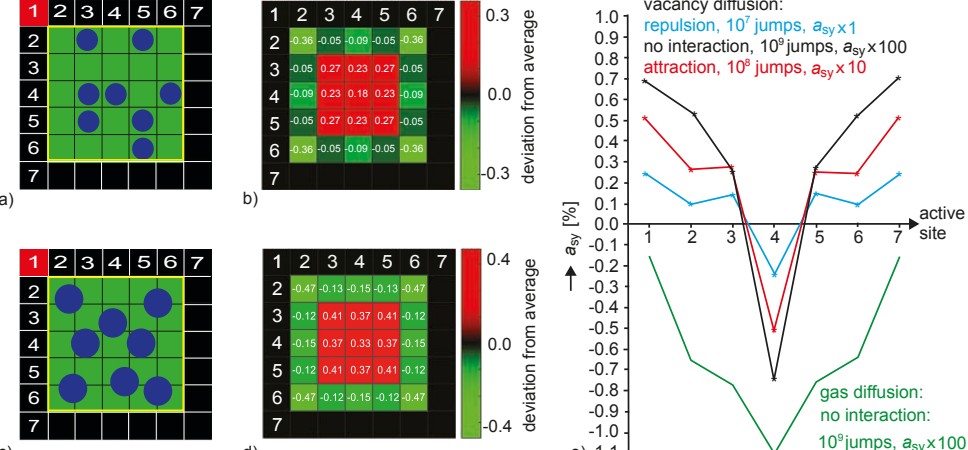


Figure 6. Population density distributions and dependences of the asymmetry
parameter $a_{\mathrm{sy}}$ on the position of the active relaxation site in the pore wall for $P = 0.3$
and a pore with $5 \times 5$ cells. a,b): Vacancy diffusion at $T = 0.2$ without particle interaction





as an average over $10^9$ jumps. a) The positions of the relaxation center in the pore wall
are numbered from 1 to 7. c,d) Gas diffusion in a square pore with cell walls 5 particle
diameters wide. a,c) Schematic drawings of the populated pores. b,d) Deviations of
the average population-density distribution from the mean. e) Variations of the
asymmetry parameter with the position of the active site in the cell wall for differently
interacting particles for vacancy and for gas diffusion. The mirror symmetry for each
trace confirms the sufficient precision of the simulation.

**4. Discussion**
For two-dimensional vacancy diffusion, jumps of randomly selected particles from one
cell to an empty neighbor cell and for two-dimensional gas diffusion with variable paths
between collisions in model pores give rise to asymmetry in the exchange statistics
between three sites depending on temperature and pressure. The two pore geometries
were investigated, one modeling a section of a porous solid grain particle (cf.
supplementary information) and the other being a small square pore with an active
site, e. g. a catalytically active center. The interaction between the particles as well as
the particles and the walls are identical in the reported data but can be repulsive, zero,
or attractive in the vacancy-diffusion case while it is zero for the gas-diffusion case.
The observed asymmetry parameters vary in a range on the order of $-1\% < a_{\mathrm{sy}} < 1\%$,
i. e., up to 1% of all particles in the pore may not follow detailed balance between all
pairs of sites but move coherently in circles between the three sites. It is emphasized
that this circular exchange is between the pools of particles representing the three
sites, and it is not a motion followed by individual particles completing circular jumps.
If the free interaction energy is set to zero in the vacancy-diffusion simulations, the
magnitude of the observed asymmetry $a_{\mathrm{sy}}$ is smaller than 0.01% (Fig. 4c,f), a value
which could only be observed at $10^9$ jumps of 7 particles in the pore for vacancy as
well as gas diffusion (Fig. 6e). Given positive or negative interaction energies, the
variations of $a_{\mathrm{sy}}$ with temperature $T$ appear rapid, reminiscent of phase transitions
(Figs. 4a Figs. S3a). The variations of $a_{\mathrm{sy}}$ with pressure corresponding to population
density $P$ are smooth (Figs. 4b, Figs. S3b). Either positive or negative values of $a_{\mathrm{sy}}$ are
observed as $T$ or $P$ change. A sign change of $a_{\mathrm{sy}}$ reports a change in the sense of the
circular exchange (cf. Fig. 1).
For a simple square pore, the asymmetry parameter varies with the position of
the active site in the cell wall, exhibiting mirror symmetry with respect to the wall center
(Fig. 6e). Moreover, the autocorrelations functions and their Fourier transforms have
been determined for the occupancy time tracks of selected cells at specific positions





inside a small square pore for $10^7$ jumps of all particles in the pore (Fig. 5). The track
function had been devised to have zero mean for the average cell population.
Depending on the position of the cell inside the pore, the autocorrelations functions
and their Fourier transforms vary. Specifically, the autocorrelation function can exhibit
a significant constant offset. At these positions inside the pore, the particle densities
are different from the pore average, and the cell is on average more empty or occupied
than expected if the exchange between all cells were the same. This conclusion is
supported by the observed deviations of the cell occupancies from the pore average
(Figs. 6b,d, Fig. S4). Near the pore wall the average population density is depleted and
varies in an oscillatory manner along the pore wall. Further towards the center of the
pore the average population density increases sharply and then tapers off towards the
pore center to a value slightly above the average density.
The observations for vacancy diffusion in a square pore with 5 × 5 cells are
compared to independent simulations of gas diffusion of non-interacting particles in a
square pore with an edge length of 5 particle diameters also allowing 7 relaxation
centers along the pore wall (Figs. 6a,c). A similar variation of the asymmetry parameter
is found as for vacancy diffusion, but the asymmetry parameter is negative for all
positions of the active site (Fig. 6e). Moreover, the depletion of the average particle
density at the pore wall and its subsequent variation towards the center are similar with
the exception, that oscillations of the average particle density along the pore wall are
weaker for gas diffusion at short observation intervals (Figs. 6b,d) but increase with
the duration of the observation intervals (Fig. S5). The lack of a sign change in the
asymmetry parameter with changing position of the active site may be explained by
destructive interference of particle collisions from multiple sites with the wall within one
discrete particle diameter and the fact, that the free path length between collisions in
gas diffusion is not limited to the next cell as in vacancy diffusion but can range up to
the pore diameter.
Taken together, the observed asymmetry in the three-site exchange and the
variation of the jump statistics with position inside the pore point at diffusive resonance
phenomena like standing waves of air in pipes as reported by Kundt (Kundt, 1866) or
of vibrating plates as reported by Chladni (Chladni, 1787). Stochastic resonance
phenomena have been observed with NMR first by Sleator, Hahn et al. (Sleator et al.,
1985) and subsequently studied in detail by Müller, Jerschow, et al. in different
scenarios (Müller and Jerschow, 2005; Schlagnitweit and Müller, 2012). In NMR, the





magnetization fluctuating with the thermal motion of the nuclear spins assumes the
role of the particles and the resonance circuit assumes the role of the pore.

Three-site exchange can be viewed as a finite difference approximation to the

Laplace operator (van Kampen, 1992; Kuprov, 2022) governing Fick's second law
(Fick, 1855). Considering some local site $N$ with neighbor sites $N$-1 and $N$+1 right and
left, the mass flow to and from site $N$ given by eqn. (1) is
$$\frac{\mathrm{d}m_N(t)}{\mathrm{d}t} = k_{N,N-1}m_{N-1} - k_{N-1,N}m_N + k_{N,N+1}m_{N+1} - k_{N+1,N}m_N, \tag{10}$$
Taking the limit to infinitesimal small distance $\Delta r \to \partial r$ between the neighboring sites
leads to $k_{j,i} = k$, demonstrating that (10) is a finite difference approximation of a
second spatial derivative balanced by the temporal variations of $m$ during infinitesimal
time $\partial t$,
$$(k\,m_{N-1} - 2\,k\,m_N + k\,m_{N+1})/\Delta r^2 \approx k\frac{\mathrm{d}^2 m}{\mathrm{d}^2 r^2} = \frac{\mathrm{d}m}{\mathrm{d}t}/\Delta r^2. \tag{11}$$
In this limit, eqn. (11) becomes Fick's second law with the diffusion coefficient $D =$
$k\Delta r^2$. This back-of-the-envelope argument indicates that the observed asymmetry of
three-site exchange is a property of Fick's second law.

The diffusion equation applicable to longitudinal magnetization in NMR instead of

particle masses $m$ is the Bloch-Torrey equation (Torrey, 1956),
$$\frac{\partial}{\partial t} m(\boldsymbol{r}, t) = D\boldsymbol{\nabla}^2 m(\boldsymbol{r}, t) - \mu\, m(\boldsymbol{r}, t), \tag{12}$$
where $m$ now is the magnetization deviation from thermal equilibrium and $\mu$ is the bulk
relaxation rate. $m(\boldsymbol{r}, t)$ solves this equation in terms of an expansion into normalized
eigenfunctions $\phi_n(\boldsymbol{r})$ with amplitudes $A_n$ and eigenvalues $\tau_n$ (Brownstein and Tarr,
1977; Song, 2000)
$$m(\boldsymbol{r}, t) = e^{-\mu t}\sum_{n=0}^{\infty} A_n\, \phi_n(\boldsymbol{r})\, e^{-\frac{t}{\tau_n}}. \tag{13}$$
The eigenvalues are determined by the boundary condition
$$D\,\boldsymbol{n}\,\boldsymbol{\nabla}\phi_n(\boldsymbol{r}) = \rho\,\phi_n(\boldsymbol{r}), \tag{14}$$
where $\rho$ is the surface relaxivity and $\boldsymbol{n}$ is the unit vector normal to the surface. The
lowest normal mode $\phi_0$ has no nodes. The higher normal modes $\phi_n$ possess nodal
surfaces. The higher diffusion eigenmodes have been detected by NMR with selective
excitation of partial pore volumes making use of field gradients internal to the pore
(Song, 2000). These experimental results reported by Song agree with the Monte Carlo
simulations of diffusive translational motion in pores reported here, in that the
population density varies across the pore and that the offset of the autocorrelation
function of the local pore occupancy depends on the position of the cell in the pore.





From the exchange asymmetry of the particles in the square pore investigated in
Fig. 6 (Fig. 7) a suggestive picture emerges for confined vacancy diffusion, where the
diffusion lengths are confined to the distances from the particle to the direct neighbor
cells. Depending on the sign of the asymmetry parameter (Fig. 7a), a small fraction of
the particles (blue circles) prefers the direct path towards or away from the active site
(red square) at the pore boundary over the path along the boundary to or from the
active site. In the center of the wall, the direct path away from the active site to the bulk
is preferred over the path along the pore wall when leaving the contact region with the
active site (Fig. 7b). But because jumps are allowed in vacancy diffusion only to
neighboring cells, the cells 2 at the wall right and left of the active site 3 must be
populated from the bulk 1 by direct jumps form the bulk to the wall. For these jumps,
the asymmetry parameter is positive, as observed for the off-center positions of the
active site (Fig. 6e). Given the symmetry of the square pore, the in-plane translational
diffusion paths resulting from the variation of the asymmetry parameters with the
position of the active site on the pore wall (Fig. 6e) demand the existence of eight
diffusion vortices inside the pore (Fig. 7d). The symmetry of this in-plane translational
diffusion pattern matches the symmetry of one of the node patterns of the out-of-plane
vibrational modes of a square plate observed by Chladni (Fig. 7e) about a quarter of a
millennium earlier (Chladni, 1787). This suggests that the dynamic of vacancy diffusion
observed in the computer model reported here is a resonance feature of the pore and
thus an eigenmode of pore diffusion. This resonance effect is far less pronounced for
gas diffusion (Fig. 7c) where the free paths between collisions can span the entire cell.
Because the mass flow from relaxation site 2 to the active site 3 can be sustained from
any position at the pore wall the asymmetry parameter does not need to change sign
when the active site moves along the pore wall (Fig. 7e), and the circular paths can
have various shapes and can extend across the entire pore, so that the vortex pattern
is largely washed out.
Given the technological importance of fluid motion in small pores in
heterogeneous catalysis (Kärger et al., 2012), it will be interesting to explore, if
correlated motion resulting from standing particle-concentration wave patterns near
pore walls can be enhanced by external drivers like ultrasound, electric or magnetic
fields. The standing waves could be enhanced by tuning the driver frequency to the
pore resonance like a musician enforces resonance modes on a musical instrument
when playing. To enhance the diffusion eigenmodes, also low-power broad-band,





forced oscillations can be considered such as in Fourier transform infrared
spectroscopy (Michelson, 1903) and stochastic NMR spectroscopy (Ernst, 1970),
while triggering free oscillations by high-power impulses may destroy the porous
medium under study.

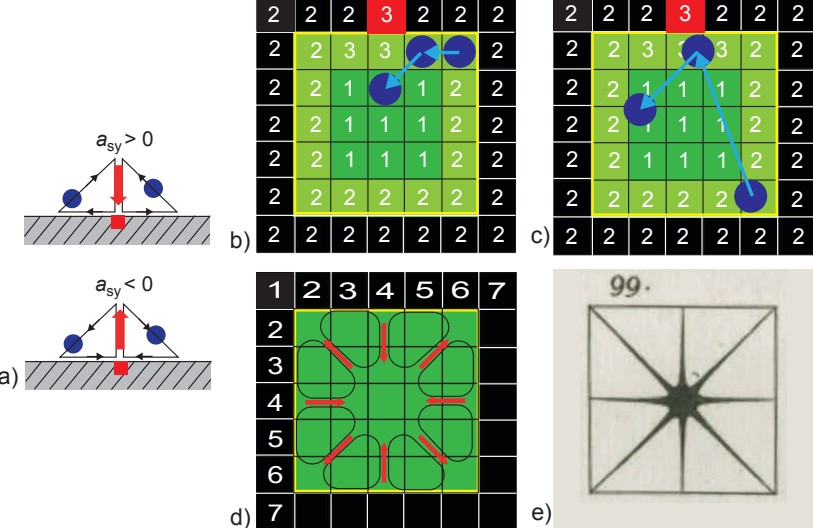


Figure 7. Illustration of the exchange asymmetry for the square pore of Fig. 6a.
a) Depending on the sign of the asymmetry parameter, a small fraction of diffusing
particles (blue circles) prefers the direct path towards or away from the active site (red
square) at the pore boundary over the path along the boundary from the active site.
b) Vacancy diffusion for negative asymmetry parameter and the active site 3 in the
center of the pore wall. Jumps are limited to the next nearest cells. The direct path
away from the active site to the bulk 1 in the center is preferred over the path along
the pore wall 2 when leaving the contact region with the active site. c) Gas diffusion for
negative asymmetry parameter and the active site 3 in the center of the pore wall. The
free paths between collisions can span the entire cell. d) In-plane translational diffusion
paths resulting from the variation of the asymmetry parameters with the position of the
active site on the pore wall depicted in Fig. 6e. e) Out-of-plane vibrational mode of a
square plate observed by Chladni (Chladni, 1787).

**5. Summary**
The evidence provided by molecular dynamics simulations of random particle jumps
on a 2D checkerboard and by simulations of 2D gas diffusion with topological
confinements supports the notion, that asymmetry in three-site exchange maps reports
the non-Brownian diffusion dynamics of confined particles. Depending on the sign of
the asymmetry parameter, for a small fraction of diffusing particles, the direct path





towards or away from the active site at the pore boundary is preferred over the path
along the boundary to or from the active site (Fig. 7). Both, vacancy diffusion and gas
diffusion produce congruent results. Yet, the reported simulations are limited to two
dimensions, and it may be argued that the asymmetry of exchange vanishes in the
more common pores with three spatial dimensions. But two-dimensional diffusion is
not an abstract model and arises for gas atoms adsorbed to metal surfaces (Oura et
al., 2013), so that the coherent particle diffusion indicated by the non-zero asymmetry
parameter may be observed there. Meanwhile experiments are in progress (Fricke and
Reimer, 2022) investigating three-site particle exchange in different systems including
highly regular pore structures such as molecular organic frameworks (Stallmach et al.,
2006; Forse et al., 2020). Moreover, future work aims at expanding the current models
of vacancy and gas diffusion to three dimensions and different types of interactions
between neighbor particles and walls to gain further insight into the details of local
particle concentrations and thermodynamic equilibrium fluxes inside pores. But given
the congruent simulation evidence for vacancy diffusion and gas diffusion in two-
dimensional confinements it is concluded, that confined diffusion is not fully random
but exhibits local concentration gradients that can lead to flux anomaly violating the
principle of detailed balance. Especially, Boltzmann's claim, that "The walls do not
interfere, because the molecules are reflected from them like elastic balls; that is,
recede from them just like that, as if the space beyond the walls would be filled with
similarly conditioned gas" is not fulfilled for small pores as evidenced by the observed
population-density variations across pores and the known existence of diffusion
eigenmodes. Accordingly, the principle of detailed balance does not apply to a fraction
of particles close to the pore wall. On a rigorous absolute scale, this means that
apparently random motion in phase-separated environments is not fully random and
time reversal does not apply to all particles. Since we live in a phase-separated
universe, Einstein may be right after all when he wrote to Bohr on 14 December 1926
(Born, 1959): "Die Quantenmechanik ist sehr achtunggebietend. Aber eine innere
Stimme sagt mir, daß das noch nicht der wahre Jakob ist. Die Theorie liefert viel, aber
dem Geheimnis des Alten bringt sie uns kaum näher. Jedenfalls bin ich überzeugt, daß
der nicht würfelt." This statement is commonly paraphrased by "God does not play
dice". It may be interesting to find out, what role diffusion eigenmodes at interfaces
between different states of matter played accumulated over billions of years in the
evolution of live in the universe.



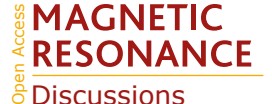


## Author Contributions

BB posed the question, executed the simulations of confined vacancy diffusion, and wrote the manuscript. MA worked out the algorithm for vacancy diffusion along with BB and supervised MP. MP programmed the algorithm for confined gas diffusion and executed the gas-diffusion simulations.

## Acknowledgement

BB thanks Thomas Wiegand at RWTH Aachen University for stimulating discussions, hosting him as a PostProf, and access to the computing facilities. He also thanks Ilya Kuprov for his flash of inspiration on linking three-site exchange to Fick's second law at a random EUROMAR 2022 breakfast encounter and Stephan Appelt, Gerd Buntkowsky, and Jeffrey Reimer for stimulating discussions. Special thanks go to the unknown reviewer who commented on the manuscript (Gao and Blümich, 2020), that asymmetry in three-site exchange violates the principle of detailed balance.

## Code availability

The codes for simulating confined 2D vacancy diffusion and confined 2D gas diffusion is available from the authors upon request.

## Conflict of Interest statement

Other than that BB is on the advisory board of Magnetic Resonance the authors declare no conflict of interest.

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
