# Peer review of "Asymmetry in Three-Site Relaxation-Exchange NMR"

_Magnetic Resonance, 2023_

## Referee Comment (RC5)

**Referee Comment on Manuscript mr-2023-8**

**Monte-Carlo Analysis of Asymmetry in Three-Site Relaxation Exchange: Probing Detailed Balance**

The authors present Monte Carlo simulations of a lattice gas using a dynamic model that breaks detailed balance. They determine a quantity called asymmetry parameter, which measures the breaking of detailed balance, and show that it is nonzero. They also present results for an off-lattice gas model which seems to behave in a similar manner. These findings are related to recent NMR experiments.

I strongly disagree with the main statements in the paper. In my opinion the work suffers from serious conceptual deficiencies regarding both the design of the model and the interpretation of the results, which is why I would absolutely not recommend it for publication in a regular journal. However, I understand that in this journal, the referee reports will be published alongside with the paper. Therefore, a publication might be acceptable as long as some additional technical issues have been fixed.

Let me start with the technical issues first

- The central quantity, the asymmetry parameter, is never properly defined. The only definition is found in Equation (5) which refers to the special case of a three-site exchange. The authors must add an equation defining the quantity which is actually measured in the simulations and shown in Figures 4 and 6.

- Likewise, a so-called "active site" seems to be an important ingredient either of the dynamical model or in the analysis (this does not become clear), but it is never defined. It has "different relaxation properties" but relaxation properties have not been introduced in the definition of the model before. As an "explanation", the caption of Figure 3 offers the following cryptic sentence: "If a particle cell contacts two different relaxation sites, the higher number overrides the lower number when identifying its relaxation environment." What does this mean in practice? Does the presence of an active site change the dynamics or is it just important for the analysis? And how exactly is this implemented?

- Apart from the active site element, I think I roughly understand the dynamical model of the lattice simulations, but the off-lattice simulations (Section 2.2) are not well explained at all. Simulations of hard particle models would typically be done using event-driven algorithms, where the system is propagated from one elastic collision to the next. Apparently, this was not done here, instead fixed time steps were used, which reduces the accuracy of the simulations. How exactly were the collisions implemented? For example, did the authors accurately account for the

impact parameter of each collision when calculating the new momenta of the participating particles, or did they pick them at random? What was the length of the time step? How did they handle situations when three particles collide within one time step? Such information is crucial if you report on simulation results that supposedly break the second law of thermodynamics.

- Error bars are missing throughout. They must be added in the graphs, also the numbers in the text should be given with errors, especially those for (nonzero) asymmetry parameters.

- Given the complexity of the model, the code should not just be "available upon request", it should be published together with the manuscript. This holds especially for the off-lattice code.

These issues must be fixed before the paper can be published.

Next I will summarize the conceptual deficiencies in the presentation of the paper.

**Monte Carlo model** :

**Description of the model** :

*Helmholtz free energy*: On page 7, it is claimed that "the particle motion is governed by the Helmholtz free energy $A$". However, the Helmholtz free energy is a global thermodynamic quantity and does not govern local microscopic dynamics. Probably, the authors to refer to some kind of effective coarse-grained potential here.

*Dynamics and Boltzmann distribution*: Same page, the authors state "The probability of a particle moving from one cell to another is given by the Boltzmann distribution $p = \exp(-\Delta A/k_B T)$". This statement does not make sense, as already apparent from the fact that the "probability" $p$ can be larger than one, $p > 1$ for $\Delta A < 1$. It is also not consistent with the subsequent description of the algorithm, where it becomes clear that the probability of moving to a certain site also depends on the number of equivalent accessible sites etc.

**Design of the model** :

*Internal energy*: The internal energy change after moving one particle is described as $\Delta U = \mathbf{F} \Delta \mathbf{R}$ (page 7), where $\mathbf{F}$ is the a force acting on a particle that is constructed from the occupancy of neighboring sites. First, there is an obvious sign error there, probably a typo, it should really read $\Delta U = -\mathbf{F} \Delta \mathbf{R}$: The energy decreases if the particle follows the force. For example, in a gravitational field, if you roll downhill, your potential energy decreases. Second, and more seriously, it is easy to see that this specific force field, as it is formulated on a lattice, is

not conservative. For example, consider a system where one particle is fixed at the origin, and a second particle undergoes a cyclic motion from $(1,0) \rightarrow (2,0) \rightarrow (2,1) \rightarrow (1,1) \rightarrow (1,0)$. Then the total internal energy change after the cycle is not zero, even though the final and initial configuration are exactly the same. Therefore, this lattice force field cannot be derived from a potential.

*Entropy*: The probability of moving to a neighbor lattice site is associated with an entropy change, which is estimated by the sum of step lengths to unoccupied neighbor cells. This specific form of entropy is entirely heuristic and again, it cannot be derived from an effective entropy potential. One should also note that it is not necessary to include translational entropy in a proper Monte Carlo algorithm: The algorithm will automatically account for it.

*Jump probability* (page 8): From the previous two points, it is already clear that the quantity $\Delta A$ in the expression for $p$ cannot be associated with a well-defined effective potential $A$. However, even if such a potential existed, the choice of jump probabilities seems rather arbitrary. For example, page 8 says "If $0 < p < 1$, the destination cell is chosen at random from all those with the same largest jump probability $p < 1$". This is not well motivated. Why not choose from all cells with weighted probabilities according to their jump probability?

The algorithm described here is not motivated by any microscopic considerations. With the same right, assuming that $\Delta A$ could really be derived from a global effective potential function $A$, one could also use a standard Metropolis algorithm, which would satisfy detailed balance by construction.

*Summary*: The presented Monte Carlo algorithm does not satisfy detailed balance for two reasons: First, even though the notation suggests otherwise, the underlying quantities $p$ are not associated with a well-defined effective energy function $A$. Second, the jump probabilities are chosen heuristically according to some random rules which are not well-motivated. It is not surprising that these rules do not satisfy detailed balance, because imposing detailed balance usually requires special efforts.

In fact, these rules would not even guarantee global balance if $A$ were well-defined. On the other hand, they do define some kind of stochastic Markovian dynamics, and according to the central limit theorem of finite Markov systems, the probability distribution will converge against some stationary fixed point, which however differs from the Boltzmann distribution $\mathcal{N} \exp(-\beta A)$. Furthermore, this stationary state would include persistent currents by default, because, as explained above, special efforts must be taken to remove them in such a model.

**Interpretation of the results** :

*Thermodynamic equilibrium*: The term "thermodynamic equilibrium", by definition, refers to a stationary state *without currents*. One of the central postulates of thermodynamics is that every physical closed dynamical system reaches thermodynamic equilibrium at some point. This is a postulate and might be debated. However, a system with persistent currents as described in the manuscript would not be considered to be at thermodynamic equilibrium.

*Detailed balance and nonequilibrium thermodynamics:* As correctly stated in the manuscript, the lack of currents is associated with microscopic detailed balance – or, putting it the other way round, breaking detailed balance normally generates currents. However, this also implies that entropy is constantly being produced, and dissipated, see, e.g., References [1-3].

Dynamical systems with broken detailed balance have been discussed in nonequilibrium thermodynamics for many decades. Physically, they are used to describe open dissipative systems, for example, living systems or active systems [1,2], which are stabilized via a steady input of energy. It is easily possible to design stochastic dynamical systems that break detailed balance, as has been done, e.g., in the present manuscript or in Refs. [3,4]. In Monte Carlo simulations, implementing such dynamics can have the advantage that a desired probability distribution function can be sampled much more efficiently [4].

*Detailed balance and Monte Carlo:* The Monte Carlo method has been introduced by Metropolis et al as a method to efficiently sample a desired target probability distribution. The necessary ingredient for this is to impose global balance. Detailed balance is not strictly necessary. With the exception of kinetic Monte Carlo (which has not been used here), Monte Carlo dynamics is typically not realistic.

Nevertheless, Monte Carlo is also used to study dynamical systems in a coarse-grained sense. However, it is important to note that in this type of model, you get out what you put in. If you implement Monte Carlo moves that break detailed balance, then clearly, you will find that detailed balance is broken in your system. Therefore, Monte Carlo simulations designed to model dynamics at thermal equilibrium must be set up such that the Monte Carlo moves satisfy detailed balance.

*Is detailed balance always fulfilled?* As stated above, the claim that closed physical dynamical systems reach thermodynamic equilibrium is a postulate. It lies at the heart of the second law of thermodynamics, but being a postulate, it could be violated in certain cases. In fact, it is violated, e.g., for integrable systems such as linear harmonic

chains. It has not been proved rigorously except for a few special cases.

On the other hand, the opposite claim that detailed balance might be broken in realistic (closed) physical system fundamentally challenges the foundations of thermodynamics. Such a claim cannot be based on Monte Carlo simulations. This is because, as explained above, Monte Carlo dynamics are inherently artificial, and it is much easier to implement dynamical models that break detailed balance than to implement models that satisfy detailed balance. The claim would have to be based on experiments, or on molecular simulations of a truly microscopic model, e.g., classical Hamiltonian dynamics or Schrödinger dynamics. In fact, there have been several claims in the past, based on atomistic simulations, that the second law might be broken in nanoscale systems. For example, spontaneous unidirectional currents through pores or the like were observed in simulations. In all of these cases, it eventually turned out that the claimed effects could be attributed to numerical artefacts of the simulations.

The central question is whether a system can thermalize, which is a valid question especially for nanoscale systems and subject of active research. Specifically, the gas diffusion case discussed in the manuscript is related to the question whether a classical ideal gas can thermalize. This is one of the few cases which has been studied very intensely and for which rigorous results exist (the H-theorem, see [5]). Ideal gases do thermalize! In the manuscript, nonideal gases with excluded volume interactions are considered, which might change the situation, but I would be very surprised if it did. This is one of the reasons why it is so important that the authors describe their simulations for the gas diffusion simulations in more detail. If they maintain the claim that detailed balance is broken in these (off-lattice) systems, they should prove it much more carefully, e.g., by systematic variation of the time step, by studying the relaxation of several quantities as a funciton of simulation time, and by a solid assessment of error bars.

References:

**1** C. W. Lynn et al, PNAS 2021, 118, e2108998118.

**2** F. S. Gnesotto et al, Rep. Prog. Phys. 2018, 81, 066601.

**3** L. Crochik et al, Phys. Rev. E 2005, 72, 057103.

**4** M. Michel et al, J. Chem. Phys. 2014, 140, 054116.

**5** G. Truesdell, R.G. Muncaster, *Fundamentals of Maxwell's kinetic theory of a simple monatomic gas*, Chapter XI, in *Pure and Applied Mathematics*, Volume 83, pp. 145-172 (1980).

---

## Author Comment (AC8)

**Reply to reviewer #2**

A  The authors (A) honestly thank reviewer #2 (R) for his/her critical reading, his/her extensive comments, and the time and effort spent with reviewing the manuscript.

R  The authors present Monte Carlo simulations of a lattice gas using a dynamic model that breaks detailed balance. They determine a quantity called asymmetry parameter, which measures the breaking of detailed balance, and show that it is nonzero. They also present results for an off-lattice gas model which seems to behave in a similar manner. These findings are related to recent NMR experiments.

   I strongly disagree with the main statements in the paper. In my opinion the work suffers from serious conceptual deficiencies regarding both the design of the model and the interpretation of the results, which is why I would absolutely not recommend it for publication in a regular journal. However, I understand that in this journal, the referee reports will be published alongside with the paper. Therefore, a publication might be acceptable as long as some additional technical issues have been fixed.

   Let me start with the technical issues first

R  The central quantity, the asymmetry parameter, is never properly defined. The only definition is found in Equation (5) which refers to the special case of a three-site exchange. The authors must add an equation defining the quantity which is actually measured in the simulations and shown in Figures 4 and 6.

A  The quantity reported in Figs. 4 and 6 is the very quantity defined in Eqn. (5). It follows from the master Eqn. (1) for the special case of three sites at mass equilibrium, when the time derivative on the left-hand side of (1) vanishes. The general case for $n$-site exchange as described in the van Kampen text can be worked out accordingly by introducing general indices $(k_{ij}M_j - k_{ji}M_i)$. $k_{ij}M_j$ quantifies the number of particles moving from site $j$ to site $i$ per time. This number corresponds to the peak integral in a $T_2$-$T_2$ NMR exchange map after correction for the impact of NMR relaxation. Thus, it is a quantity that can be determined experimentally with the help of forward modelling and inverse Laplace transformation (see Eqn. 7). Note, that $T_2$ is a relaxation time for nuclear magnetization which describes the decay of the impulse response measured in NMR experiments. The denominator of Eqn. (5) is the sum over all particle moves, so that Eqn. (5) specifies the imbalance of the fractions of particles that move from pool $j$ to pool $i$ versus those that move from pool $i$ to pool $j$. Most readers of this journal are familiar with the $T_2$-$T_2$ experiment or can refresh their memory with the references given in the manuscript. In practice, a two-dimensional exchange map measured with NMR contains peaks on the diagonal and cross-peaks right and left of the diagonal. It has been a question of concern in parts of the NMR community, if such an exchange map must always be symmetric with respect to its diagonal. If detailed balance is obeyed, then the corresponding cross-peak integrals right and left of the diagonal are the same and the map is symmetric. We investigate three-site exchange as the simplest case where $T_2$-$T_2$ exchange maps could formally

become asymmetric. In this case the master equation (1) at mass balance demands that the cross-peak integrals in the three-site $T_2$-$T_2$ NMR exchange map on each side of the diagonal always obey Eqn. (4). This was critically observed in each simulation by counting particle jumps to and from three different pools. The total number of observations in the simulations was chosen so that the agreement between the three symmetry parameters following from Eqn. (4) was usually at least within two decimals.

R    Likewise, a so-called "active site" seems to be an important ingredient either of the dynamical model or in the analysis (this does not become clear), but it is never defined. It has "different relaxation properties" but relaxation properties have not been introduced in the definition of the model before. As an "explanation", the caption of Figure 3 offers the following cryptic sentence: "If a particle cell contacts two different relaxation sites, the higher number overrides the lower number when identifying its relaxation environment." What does this mean in practice? Does the presence of an active site change the dynamics or is it just important for the analysis? And how exactly is this implemented?

A    The active site is a chemical term used in the context of heterogeneous catalysis, where diffusion of molecules in and out of nanopores is essential, and somewhere in the nanopore there is a chemical site that activates the catalysis of a chemical reaction. Often the active site is highly paramagnetic, i. e. molecules in contact with the site have a short NMR relaxation time $T_2$, which differs from the NMR relaxation time $T_2$ of molecules in contact with other parts of the pore wall (intermediate $T_2$) and molecules in the bulk fluid phase (long $T_2$). This is a simplified explanation, and the NMR experts know, that the pore size and the exchange kinetics play an important role here. The presence of the active site does not impact the dynamics, but it gives a handle on identifying three distinct pools of particles diffusing inside the pore. In our simulations the membership of a diffusing particle belonging to one of three pools is identified by their relaxation time. If the particle is without contact to other particles or in contact with only particles in the bulk, then it belongs to pool 1. If it is in contact with a particle from the bulk and with the wall, it belongs to pool 2. If it is in contact with either the wall (pool 2) or the bulk (pool 1) and the active site, then it is counted to belong to pool 3. This is what the statement in the caption to Fig. 3 is meant to express, and this is how the dynamics were implemented in the program.

R    Apart from the active site element, I think I roughly understand the dynamical model of the lattice simulations, but the off-lattice simulations (Section 2.2) are not well explained at all. Simulations of hard particle models would typically be done using event-driven algorithms, where the system is propagated from one elastic collision to the next. Apparently, this was not done here, instead fixed time steps were used, which reduces the accuracy of the simulations. How exactly were the collisions implemented? For example, did the authors accurately account for the impact parameter of each collision when calculating the new momenta of the participating particles, or did they pick them at random? What was the length of

the time step? How did they handle situations when three particles collide within one time step? Such information is crucial if you report on simulation results that supposedly break the second law of thermodynamics.

A   The length of the time steps in the algorithm is in arbitrary units. The simulation was set up such that an initial speed of 0.035 corresponds to moving 0.035 arbitrary length units in 1 arbitrary time unit. This can be converted to SI units, but it is unnecessary as the focus is on the number of times the particles change relaxation sites. At each time step, the distance between each possible pair of particles was considered. If the center of each particle was within one diameter of another, the particles are considered to have collided. Immediately after a collision the projection of the velocity vector along the collision axis is reversed prior to propagating to the next step. At these low occupancy numbers, in the very rare occasion that more than two particles simultaneously collide, the projection of each particle's velocity vector on the collision axis is reversed prior to propagating to the next step. The impact of the observation interval on the asymmetry parameter has been reported in Fig. S6.

R   Error bars are missing throughout. They must be added in the graphs, also the numbers in the text should be given with errors, especially those for (nonzero) asymmetry parameters.

A   Thank you for making us aware of the missing error discussion. We apologize for not having noticed it disappear in the process of revising our draft manuscript multiple times before submission. In the submitted manuscript the simulation error has only indirectly been addressed in line 421, where the mirror symmetry of the observed asymmetry parameter with the position of the active site in the pore wall is pointed out. Also following the comments of reviewer #1, we will add a statement, that the total number of observations in the simulations was chosen so that the agreement between the three signed symmetry parameters stated in Eqn. (4) was usually met within at least two decimals. Thus, error bars in the graphs would be too small to be visible.

R   Given the complexity of the model, the code should not just be "available upon request", it should be published together with the manuscript. This holds especially for the off-lattice code.

These issues must be fixed before the paper can be published.

A   We will include both codes in the supplement.

Next I will summarize the conceptual deficiencies in the presentation of the paper.

**Monte Carlo model** :

  **Description of the model** :

R    *Helmholtz free energy*: On page 7, it is claimed that "the particle motion is governed by the Helmholtz free energy $A$". However, the Helmholtz free energy is a global thermodynamic quantity and does not govern local microscopic dynamics. Probably, the authors to refer to some kind of effective coarse-grained potential here.

     *Dynamics and Boltzmann distribution*: Same page, the authors state "The probability of a particle moving from one cell to another is given by the Boltzmann distribution $p = \exp(-\Delta A/k_B T)$". This statement does not make sense, as already apparent from the fact that the "probability" $p$ can be larger than one, $p > 1$ for $\Delta A < 1$. It is also not consistent with the subsequent description of the algorithm, where it becomes clear that the probability of moving to a certain site also depends on the number of equivalent accessible sites etc.

A    This point has been addressed in line 220: "If for one or more jumps $p \geq 1$, the destination cell of the jump is picked at random from this subset of all potential jumps."

R    **Design of the model** :

     *Internal energy*: The internal energy change after moving one particle is described as $\Delta U = \mathbf{F}\Delta\mathbf{R}$ (page 7), where $\mathbf{F}$ is the a force acting on a particle that is constructed from the occupancy of neighboring sites. First, there is an obvious sign error there, probably a typo, it should really read $\Delta U = -\mathbf{F}\Delta\mathbf{R}$: The energy decreases if the particle follows the force. For example, in a gravitational field, if you roll downhill, your potential energy decreases.

A    Thank you for pointing out the sign error in $\Delta U = -\mathbf{F}\ \Delta\mathbf{R}$. We will correct that. Instead of the symmetric expression $\Delta U = -(\mathbf{F}_{\text{final}}-\mathbf{F}_{\text{initial}})\Delta\mathbf{R}$ we have chosen the computationally faster asymmetric difference neglecting $\mathbf{F}_{\text{final}}$ obtaining $\Delta U = \mathbf{F}_{\text{Initial}}\Delta\mathbf{R}$.

R    Second, and more seriously, it is easy to see that this specific force field, as it is formulated on a lattice, is not conservative. For example, consider a system where one particle is fixed at the origin, and a second particle undergoes a cyclic motion from $(1,0) \rightarrow (2,0) \rightarrow (2,1) \rightarrow (1,1) \rightarrow (1,0)$. Then the total internal energy change after the cycle is not zero, even though the final and initial configuration are exactly the same. Therefore, this lattice force field cannot be derived from a potential.

A    This is an interesting point. So, on average, energy can be injected into the system at each jump, which may give rise to the observed imbalance (asymmetry) of the three-site exchange. On the other hand, we observed the asymmetry also without prioritizing jumps based on probabilities calculated from our crude approximations of internal energy and entropy in the vacancy-diffusion case. This is why we checked our calculations with a different scenario, i. e. for confined gas diffusion of spheres colliding elastically and starting with an arbitrary velocity distribution. This distribution thermalized rapidly and produced asymmetry parameters in accordance with vacancy diffusion (Fig. 6e). Therefore,

if energy is injected onto the system, then it must be the case for both scenarios, vacancy diffusion and gas diffusion, in which particles diffuse randomly without thermodynamic constraints. So far, we cannot see where in both algorithms that can happen.

R    *Entropy*: The probability of moving to a neighbor lattice site is associated with an entropy change, which is estimated by the sum of step lengths to unoccupied neighbor cells. This specific form of entropy is entirely heuristic and again, it cannot be derived from an effective entropy potential. One should also note that it is not necessary to include translational entropy in a proper Monte Carlo algorithm: The algorithm will automatically account for it.

A    Thank you for your clarifying remark. Yes, the definition of entropy is entirely heuristic and only has essential features of the statistical entropy. Please also see our response to first reviewer: Our use of these crude approximations for $S$ and $\mathbf{F}$ turned out to be important for discovering the nonzero asymmetry parameter in the vacancy-diffusion simulations in the first place as it resulted in a large asymmetry parameter. The gas-diffusion simulations were conducted without consideration of any interaction energy other than elastic collisions between spheres. Following these simulations, we executed the vacancy diffusion simulations again but now with $\Delta U=0=\Delta S$ and at 100 times higher number of jumps. Only then we discovered that there is still nonzero asymmetry albeit about one to two orders of magnitude lower (Fig. 6e, black curve). Our crude approximations of $\Delta U$ and $\Delta S$ for the calculation of jump probabilities do not impact the conclusion of nonzero asymmetry parameter, but the use of jump probabilities is discussed in the manuscript, because it was decisive in the discovery process of the computer simulation experiments. We will state that in the revised manuscript.

R    *Jump probability* (page 8): From the previous two points, it is already clear that the quantity $\Delta A$ in the expression for $p$ cannot be associated with a well-defined effective potential $A$. However, even if such a potential existed, the choice of jump probabilities seems rather arbitrary. For example, page 8 says "If $0 < p < 1$, the destination cell is chosen at random from all those with the same largest jump probability $p < 1$". This is not well motivated. Why not choose from all cells with weighted probabilities according to their jump probability?

A    If the jump probability turned out to be larger than one it was artificially set to 1. Also, on occasion it happened that the largest jump probabilities were smaller than 1 and that the largest two or more probabilities were equal. In those two cases the destination cell was picked at random from the respective subsets to proceed in an unbiased way.

R    The algorithm described here is not motivated by any microscopic considerations. With the same right, assuming that $\Delta A$ could really be derived from a global effective potential function $A$, one could also use a standard Metropolis algorithm, which would satisfy detailed balance by construction.

A    Choosing an algorithm that satisfies detailed balance by construction precludes testing detailed balance.

R  *Summary*: The presented Monte Carlo algorithm does not satisfy detailed balance for two reasons: First, even though the notation suggests otherwise, the underlying quantities *p* are not associated with a well defined effective energy function *A*. Second, the jump probabilities are chosen heuristically according to some random rules which are not well-motivated. It is not surprising that these rules do not satisfy detailed balance, because imposing detailed balance usually requires special efforts.

   In fact, these rules would not even guarantee global balance if *A* were well-defined. On the other hand, they do define some kind of stochastic Markovian dynamics, and according to the central limit theorem of finite Markov systems, the probability distribution will converge against some stationary fixed point, which however differs from the Boltzmann distribution N exp($-\beta A$). Furthermore, this stationary state would include persistent currents by default, because, as explained above, special efforts must be taken to remove them in such a model.

A  We repeat, that our choice of *p* does not impact our main conclusion of nonzero asymmetry parameter, but it is mentioned in the manuscript, because it was important in the discovery process of the computer simulation experiments. We found nonzero asymmetry parameters also when choosing the destination cell for a jump from all vacant neighbor cells at random.

**Interpretation of the results** :

R  *Thermodynamic equilibrium*: The term "thermodynamic equilibrium", by definition, refers to a stationary state *without currents*. One of the central postulates of thermodynamics is that every physical closed dynamical system reaches thermodynamic equilibrium at some point. This is a postulate and might be debated. However, a system with persistent currents as described in the manuscript would not be considered to be at thermodynamic equilibrium.

A  Perhaps the wording of current is misleading. The result of the simulations is that there is a cyclic exchange between random particles from one of three pools and the next pool when this movement happens under the impact of topological constraints. Our simulations suggest that this step is not balanced in detail on the particle scale, i. e. on the scale a few particle diameters from the pore wall. This means, that depending on the sign of the asymmetry parameter, a particle may jump with slight preference directly from the wall to the bulk at certain positions near the wall as compared to jumping form the bulk to the wall. Instead, the particles arriving at these special positions on the wall prefer to first move along the wall before jumping back to the bulk. These positions of a large asymmetry parameter depend on the geometry of the confinement. This is an explanation at the molecular level. The notion of currents refers to a macroscopic picture. We will check our manuscript if we have used the right wording, made this point sufficiently clear, and correct where necessary.

R  *Detailed balance and nonequilibrium thermodynamics:* As correctly stated in the manuscript, the lack of currents is associated with microscopic detailed balance – or, putting it the other way round, breaking detailed balance normally generates

currents. However, this also implies that entropy is constantly being produced, and dissipated, see, e.g., References [1-3].

A    See previous response and thank you for the literature references. In view of this issue, we refer to Feynman's rachet in the manuscript at line 154.

R    Dynamical systems with broken detailed balance have been discussed in nonequilibrium thermodynamics for many decades. Physically, they are used to describe open dissipative systems, for example, living systems or active systems [1,2], which are stabilized via a steady input of energy. It is easily possible to design stochastic dynamical systems that break detailed balance, as has been done, e.g., in the present manuscript or in Refs. [3,4]. In Monte Carlo simulations, implementing such dynamics can have the advantage that a desired probability distribution function can be sampled much more efficiently [4].

A    Thank you for these further references. Indeed, there are other steady-state cases that can be mentioned. We refer to some in the introduction, lines 40-44.

R    *Detailed balance and Monte Carlo:* The Monte Carlo method has been introduced by Metropolis et al as a method to efficiently sample a desired target probability distribution. The necessary ingredient for this is to impose global balance. Detailed balance is not strictly necessary. With the exception of kinetic Monte Carlo (which has not been used here), Monte Carlo dynamics is typically not realistic.

Nevertheless, Monte Carlo is also used to study dynamical systems in a coarse-grained sense. However, it is important to note that in this type of model, you get out what you put in. If you implement Monte Carlo moves that break detailed balance, then clearly, you will find that detailed balance is broken in your system. Therefore, Monte Carlo simulations designed to model dynamics at thermal equilibrium must be set up such that the Monte Carlo moves satisfy detailed balance.

*Is detailed balance always fulfilled?* As stated above, the claim that closed physical dynamical systems reach thermodynamic equilibrium is a postulate. It lies at the heart of the second law of thermodynamics, but being a postulate, it could be violated in certain cases. In fact, it is violated, e.g., for integrable systems such as linear harmonic chains. It has not been proved rigorously except for a few special cases.

On the other hand, the opposite claim that detailed balance might be broken in realistic (closed) physical system fundamentally challenges the foundations of thermodynamics. Such a claim cannot be based on Monte Carlo simulations. This is because, as explained above, Monte Carlo dynamics are inherently artificial, and it is much easier to implement dynamical models that break detailed balance than to implement models that satisfy detailed balance. The claim would have to be based on experiments, or on molecular simulations of a truly microscopic model, e.g., classical Hamiltonian dynamics or Schr¨odinger dynamics. In fact, there have been several claims in the past, based on atomistic simulations, that the second law might be broken in nanoscale systems. For example, spontaneous unidirectional currents through pores or the like were observed in

simulations. In all of these cases, it eventually turned out that the claimed effects could be attributed to numerical artefacts of the simulations.

A  We understand that you are saying that Monte Carlo methods cannot be used for this type of study on a molecular scale. We concede, that although both algorithms produce the same results, there is no proof, that for some reason still unknown to us they both produce the same false result. This is why we are currently working on experiments in addition to the one mentioned in the introduction to corroborate or falsify the computer-simulation results, see lines 575-578 in the manuscript. In view of your comments, we will formulate our conclusions more as a question.

R  The central question is whether a system can thermalize, which is a valid question especially for nanoscale systems and subject of active research. Specifically, the gas diffusion case discussed in the manuscript is related to the question whether a classical ideal gas can thermalize. This is one of the few cases which has been studied very intensely and for which rigorous results exist (the H-theorem, see [5]). Ideal gases do thermalize! In the manuscript, nonideal gases with excluded volume interactions are considered, which might change the situation, but I would be very surprised if it did. This is one of the reasons why it is so important that the authors describe their simulations for the gas diffusion simulations in more detail. If they maintain the claim that detailed balance is broken in these (off lattice) systems, they should prove it much more carefully, e.g., by systematic variation of the time step, by studying the relaxation of several quantities as a function of simulation time, and by a solid assessment of error bars.

A  Thermalization of the gas diffusion has been tested starting from both a Maxwell distribution and from a Dirac distribution of all particles possessing the same initial speed. For both large and small pores, the velocity distributions approach Maxwell-Boltzmann which is apparent within the first 1000 steps. We will add a sentence to that end in the revised manuscript and show the approach of an initial Dirac distribution to the Maxwell-Boltzmann distribution of speeds for the gas-diffusion case at three time intervals, initial, intermediate, and final, in a revision of the supporting information.

References:

1  C. W. Lynn et al, PNAS 2021, 118, e2108998118.

2  F. S. Gnesotto et al, Rep. Prog. Phys. 2018, 81, 066601.

3  L. Crochik et al, Phys. Rev. E 2005, 72, 057103.

4  M. Michel et al, J. Chem. Phys. 2014, 140, 054116.

5  G. Truesdell, R.G. Muncaster, *Fundamentals of Maxwell's kinetic theory of a simple monatomic gas*, Chapter XI, in *Pure and Applied Mathematics*, Volume 83, pp. 145-172 (1980).

---

## Author Response (AR1)

**Letter to the editor**

Dear Prof. Corzilius, dear Björn,

Thank you for the opportunity to revise our submission benefitting from the insights of the expert reviewers. Originally, we were led to submitting the results of our Monte

Carlo simulations for publications in view of the two independent algorithms giving similar results, despite their provocative nature. Thanks to the open discussion in

Magnetic Resonance, and the expert advice of the anonymous reviewer on Monte

Carlo simulations, shortcomings in both algorithms were identified which violate the zero-energy balance in both cases and lead to the observed asymmetry of three-site exchange. The short summary is that none of the two algorithms maintained thermodynamic equilibrium but generated a dynamic or driven equilibrium, whereby overall mass balance was maintained. In NMR we are familiar with driven equilibrium situations when the spins are in equilibrium with the excitation, as for instance in CW

NMR or stochastic NMR. Concerning translational motion of molecules in pores, thermodynamic equilibrium corresponds to noise or Brownian motion, while our simulations showed, that driven equilibrium can lead to coherent circular motion in the pore. It is a question to be investigated further if such motion can be stimulated in pores by ultrasonic, electric, or magnetic fields, which might be beneficial, for example, in heterogeneous catalysis.

In view of the full discussion and the major revision of the manuscript being publicly accessible we reorganized the points of the reviewers below and answer each of them. Major changes concern the title, the introduction, and the discussion. Changes in the text are marked in yellow in the revised manuscript and explained in the following.

With kind regards,

Bernhard Blümich

**Response to reviewers**

Reviewer #1: Malcom Levitt; Reviewer #2: Anonymous; Community: Tom Barbara

The authors sincerely thank both reviewers for considerable their time and effort in dealing with our manuscript. We especially thank reviewer #2 for education us on

Monte Carlo Simulations and providing valuable literature references. We also thank

Tom Barbara for enlightening suggestions and discussions.

**Reviewer #1**

1) My first question is whether the simulations, and indeed the NMR observations
which seem to have stimulated them, are performed in equilibrium. As far as
the simulations are concerned, it is not obvious to me how one would ensure
that the simulations do correspond to an equilibrium state. One way to
establish that would be to check whether the detailed balance condition holds
- which would of course defeat much of the purpose of this study. If the
simulated system does +not+ correspond to an equilibrium system, it would
not be very surprising if detailed balance is violated in some cases. Even in
everyday life non-equilibrium states can lead to circulating flows. One might
even push this argument further and state that the results of Fig.4 etc., which
show clear violations of detailed balance (assuming that I understand the
"asymmetry" correctly), indicate that these simulations do +not+ correspond to
equilibrium. The author should at least address this possibility and its possible
implications.

Reply: Thermodynamic equilibrium and driven equilibrium need indeed be
discriminated. Driven equilibrium or overall equilibrium requires mass balance, i. e.
Eqn. (1) to be zero, and thermodynamic equilibrium requires the asymmetry
parameter, which expresses relative flux and is defined in Eqn. 5, to be zero. This point
was not clear in the original submission and is now explicitly stated in the revised
manuscript on page 3. As a result of the open discussion, it became apparent that we
are observing a driven equilibrium and not thermodynamic equilibrium in agreement
with the reviewer's objection. Accordingly, the manuscript underwent major revision.

2) The curious definition of the entropy of a site (Eq.9) puzzles me greatly. It
seems rather arbitrary, or at least its validity is not discussed. Could some of
the curious observations be linked to an entropy definition that does not satisfy
all of the necessary attributes of entropy?

To expand a little further on RC1, the common definition of entropy is that
is proportional to the log of the number of ways to realise a particular
configuration. This ensures, for example that the entropy of two independent
systems is the sum of the individual entropies (since the number of equivalent

67   configurations is the product of the numbers for the individual systems, and

68   the log of a product is a sum. I believe that $S{\sim}\ln W$ is a crucial definition of

69   entropy from which much of stat mech follows. So, I don't think it can be valid

70   to introduce an arbitrary function and call it entropy, without showing that (at

71   least in some limit or under some assumptions) the fundamental relationship

72   between S and $\ln W$ is preserved. I don't see how the authors definition, which

73   is based on the distance between cells, has any plausible relationship with

74   entropy.

75   This discussion is very interesting but cannot be resolved within this

76   discussion, in my opinion. The main point I want to make, in my role as referee,

77   is that any definition of entropy requires some indications that the definition,

78   perhaps within some assumptions or approximations, at least fulfils some of

79   the attributes of thermodynamic entropy. Otherwise it is just an arbitrary

80   function that cannot be called entropy, or used in place of entropy, and there

81   should be no expectation that such a function plays the same role as true

82   entropy (for example, increasing for an irreversible process in a closed

83   system). The literature Tom cites may be of help.

84 Reply: Entropy and internal energy were crudely modeled from the distances to

85 neighbor cells to introduce a free jump energy. This allowed us to study the asymmetry

86 parameter in dependence on temperature and pressure guiding us to the interpretation

87 of the driven motion inside the pore as a translational resonance effect. The entropy

88 model exhibits the basic features of the configurational entropy as is now explained in

89 the supplement, lines 44 to 51. We could have also approximated $W$ in $S = k_B \ln W$ by

90 the logarithm of the number of cells the particle is free to jump to. Instead, we used the

91 sum of jump distances, which for our Moore neighborhood can be argued to

92 approximate $W$ (but not the logarithm) apart from some scaling factor. This we have

93 chosen to approximate the configurational entropy for the discrete states in our

94 simulation instead of the textbook formula $S = -k_B \Sigma (P \ln P)$. In fact, we used the sum

95 of distances, because we are dealing with discrete configurations and the

96 configurations on the square grid differ, so that $S = k_B \ln W$ does not strictly apply. Our

97 choice may not be the best one, but our crude approximation exhibits the essential

98 features of entropy: The distance sum is zero, if there is only one possible

99 configuration, and it grows with the number of accessible configurations. For purpose of calculating jump probability this suffices. As subsequently, the asymmetry parameter could also be observed for jumps randomly selected from one of the free neighbor cells without resorting to internal energy or entropy, the details of the model for energy and entropy have been moved to the supplement.

**Community comment**

1) These comments are useful and worth consideration. I would like to add that these "equilibrium" conditions also play a role in describing the kinetic mechanisms for the approach to equilibrium. In that way if you have A -> B -

>C ->A cyclicly then the product of the rate constants going clockwise say, have to equal the product going counterclockwise. I believe I am correct in identifying Onsager as the origin of this notion and his classic paper is on "non equilibrium (irreversible) thermodyamics".

Reply: Thank you for referring us to Onsager's seminal work. He is now cited in line

64.

**RC5,6 Anonymous reviewer #2**

1) General comment: The authors present Monte Carlo simulations of a lattice gas using a dynamic model that breaks detailed balance. They determine a quantity called asymmetry parameter, which measures the breaking of detailed balance, and show that it is nonzero. They also present results for an off-lattice gas model which seems to behave in a similar manner. These findings are related to recent NMR experiments.

I strongly disagree with the main statements in the paper. In my opinion the work suffers from serious conceptual deficiencies regarding both the design of the model and the interpretation of the results, which is why I would absolutely not recommend it for publication in a regular journal. However, I understand that in this journal, the referee reports will be published alongside with the paper. Therefore, a publication might be acceptable as long as some additional technical issues have been fixed.

Reply: The authors are honestly grateful to the reviewer spending precious time in
educating them on Monte Carlo simulations, analyzing the submitted manuscript in
detail, and providing helpful literature references.

2)   Technical issues. These issues must be fixed before the paper can be
published.

2a)  The central quantity, the asymmetry parameter, is never properly defined. The
only definition is found in Equation (5) which refers to the special case of a
three-site exchange. The authors must add an equation defining the quantity
which is actually measured in the simulations and shown in Figures 4 and 6.

Reply: The quantity measured is exactly the quantity defined in Eqn. (5). The quantity
$M_j$ is a concentration. In the simulations it is the number of particles in pool $j$. The
quantity $k_{ij}$ denotes the rate of transitions from pool $j$ to pool $i$. The program counts the
number of particles passing from pool $j$ to $i$ and assigns that to $k_{ij} M_j$. The denominator
of (5) is the total number of jumps from one pool to another including jumps within one
pool ($k_{ii} M_i$). So, the asymmetry parameter defined in (5) is the number difference
between forward and backward jumps divided by the total number of jumps. This is the
relative circular flux. The total number of jumps is the sum of all differential jumps, i. e.
the sum over all $k_{ij} M_j$. It is calculated in the program after completion of each simulation
run and has been verified by comparison with the initially specified number of jumps.
There are no different weights assigned to particles in different pools. To better explain
the asymmetry parameter, the following text has been added (lines 73-78): "Here $k_{ij} M_j$
is the number of transitions from pool $j$ to pool $i$, corresponding to the peak integral in
an exchange map after correction for relaxation effects, so that the denominator
corresponds to the integral over all peaks. The asymmetry parameter thus quantifies
the imbalance of exchange between two sites in terms of the number of unbalanced
exchanges normalized to the total number of exchanges. Therefore, it specifies the
relative flux in the circular exchange process."

2b) Likewise, a so-called "active site" seems to be an important ingredient either
of the dynamical model or in the analysis (this does not become clear), but it
is never defined. It has "different relaxation properties" but relaxation
properties have not been introduced in the definition of the model before. As
an "explanation", the caption of Figure 3 offers the following cryptic sentence:

"If a particle cell contacts two different relaxation sites, the higher number
overrides the lower number when identifying its relaxation environment." What
does this mean in practice? Does the presence of an active site change the
dynamics or is it just important for the analysis? And how exactly is this
implemented?

Reply: The active site is a terminology used in catalysis which refers to a catalytically
active site which in the case discussed in the manuscript resides in the pore wall of a
heterogeneous catalyst. It does not change the dynamics of the particles near it, but it
typically increases their NMR relaxation rate by which the different particle pools are
identified in the $T_2$-$T_2$ relaxation-exchange NMR experiment. The numbering of the
relaxation sites is now better explained at multiple occasions: Lines 184-187: "The
NMR relaxation environments are indexed according to increasing relaxation rate. If a
particle is in contact with two different relaxation environments, it is assigned to the
relaxation environment with the higher index according to the higher relaxation rate."
Because relaxation rates are additive, this assignment is physically meaningful. Lines
268-271: "To understand the essence of the asymmetry the pore geometry was
simplified to a square with an active site in the wall to study particle motion in detail.
Particles in the bulk, in contact with the walls, and with the active site are identified by
different NMR relaxation properties (Fig. 3b)". Caption to Fig. 3 showing the two types
of pores investigated (lines 277-283): "a) Depending on their next neighbors in the first
coordination shell, the particle-relaxation environments are identified as bulk (1),
surface (2), and pore (3) with increasing relaxation rate. b) Small square pore with an
active site. The bulk (1), the walls (2), and the active site (3) have different relaxation
properties. If a particle is in contact with two different relaxation sites, it is counted to
belong to the particle pool with the larger relaxation rate, i. e. the pool with the higher
number."

2c) Apart from the active site element, I think I roughly understand the dynamical
model of the lattice simulations, but the off-lattice simulations (Section 2.2) are
not well explained at all. Simulations of hard particle models would typically be
done using event-driven algorithms, where the system is propagated from one
elastic collision to the next. Apparently, this was not done here, instead fixed
time steps were used, which reduces the accuracy of the simulations. How
exactly were the collisions implemented? For example, did the authors accurately account for the impact parameter of each collision when calculating the new momenta of the participating particles, or did they pick them at random? What was the length of the time step? How did they handle situations when three particles collide within one time step? Such information is crucial if you report on simulation results that supposedly break the second law of thermodynamics.

Reply: The description of the algorithm for the off-lattice simulations has been expanded (lines 224-251). At each time step, the distance between each possible pair of particles was considered. If the center of each particle was within one diameter of another, the particles are considered to have collided. Immediately after a collision the projection of the velocity vector along the collision axis is reversed prior to propagating to the next step, according to

$$\vec{v}_{1,\text{new}} = \vec{v}_{1,\text{old}} - \frac{2m_2}{(m_1+m_2)} \frac{\langle \vec{v}_{1,\text{old}} - \vec{v}_{2,\text{old}}, \vec{x}_1 - \vec{x}_2 \rangle}{\|\vec{x}_2 - \vec{x}_1\|^2} (\vec{x}_1 - \vec{x}_2), \tag{8}$$

$$\vec{v}_{2,\text{new}} = \vec{v}_{2,\text{old}} - \frac{2m_1}{(m_1+m_2)} \frac{\langle \vec{v}_{2,\text{old}} - \vec{v}_{1,\text{old}}, \vec{x}_2 - \vec{x}_1 \rangle}{\|\vec{x}_2 - \vec{x}_1\|^2} (\vec{x}_2 - \vec{x}_1). \tag{9}$$

At these low occupancy numbers, in the very rare occasion that more than two particles simultaneously collide, the projection of each particle's velocity vector on the collision axis is reversed prior to propagating to the next step. The length of the time steps in the algorithm is in arbitrary units. The simulation was set up such that an initial speed of 0.035 corresponds to moving 0.035 arbitrary length units in 1 arbitrary time unit.

Thanks to the reviewer's comment we now understand, that a time step orders of magnitude smaller than that should have been used. We repeated the simulations with a 100 times smaller time step and found that the asymmetry parameter decreased by a factor of about 1000, confirming the reviewer's point, that with decreasing time step the asymmetry parameter approaches zero and that the principle of detailed balance is obeyed in the limit of infinitesimally short time steps corresponding to infinitely long computation time. Nevertheless, the fact, that the three asymmetry parameters resulting from Eqn. (4) agree to within at least 2 relevant digits (lines 401–402) confirms that the particle motion reports an overall equilibrium state. Consequently, we interpret the particle motion observed with the "large" time step to be a motion not in thermodynamic equilibrium but in dynamic equilibrium driven by energy injected into the system at each collision.

2d) Error bars are missing throughout. They must be added in the graphs, also the

  numbers in the text should be given with errors, especially those for (nonzero)

  asymmetry parameters.

Reply: Error bars are not included in the graphs but are discussed in the context of Fig.

6g: "The parameter depends on the location of the relaxation center in the pore wall (Fig. 6). This dependence has been verified to be identical for all walls of the square pore. Moreover, it exhibits mirror symmetry about the center position (Fig. 6g), assuring that the simulation noise is negligible" (lines 380-384) and caption to Figure 6: "The mirror symmetry of each trace about the center position reports high precision of the simulation" (lines 374, 375). See also lines 401–402: "In all these cases the precision of the asymmetry parameter $a_{sy}$ obtained in the simulations exceeds the second relevant digit".

2e) Given the complexity of the model, the code should not just be "available upon

  request", it should be published together with the manuscript. This holds

  especially for the off-lattice code.

Reply: The codes of both algorithms are now made available in the revised supplementary material.

3)    Conceptual deficiencies in the presentation of the paper.

3a) Monte Carlo model: Description of the model:

  Helmholtz free energy: On page 7, it is claimed that "the particle motion is

  governed by the Helmholtz free energy A". However, the Helmholtz free energy

  is a global thermodynamic quantity and does not govern local microscopic

  dynamics. Probably, the authors to refer to some kind of effective coarse-

  grained potential here.

  Dynamics and Boltzmann distribution: Same page, the authors state "The

  probability of a particle moving from one cell to another is given by the

  Boltzmann distribution $p = \exp(-\Delta A/(k_B\ T))$". This statement does not make

  sense, as already apparent from the fact that the "probability" $p$ can be larger

  than one, $p > 1$ for $\Delta A < 1$. It is also not consistent with the subsequent description of the algorithm, where it becomes clear that the probability of moving to a certain site also depends on the number of equivalent accessible sites etc.

Reply (see also our response to reviewer #1): A jump probability has been introduced to allow studies as a function of temperature and pressure, which helps to understand the nature of the observed asymmetry of exchange as a resonance effect. We agree that the concept is far-fetched, and that the definition of the free energy is heuristic.

Because the asymmetry was subsequently also observed for arbitrary jumps to free neighbor cells, the description of the fee-energy model has been moved to the supplementary material. If a jump probability larger than resulted from the model it was set to 1 in the algorithm for computational purpose and the destination cell for the jump was picked at random from all destination cells with the same jump probability. This is explained in line 53–57 of the supplement, in particular: "If for one or more jumps $p \geq$

1, the destination cell of the jump is picked at random from this subset of all potential jumps."

3b) Design of the model:

Internal energy: The internal energy change after moving one particle is described as $\Delta U = \boldsymbol{F}\Delta\boldsymbol{R}$ (page 7), where $\boldsymbol{F}$ is a force acting on a particle that is constructed from the occupancy of neighboring sites. First, there is an obvious sign error there, probably a typo, it should really read $\Delta U = -\boldsymbol{F}\Delta\boldsymbol{R}$: The energy decreases if the particle follows the force. For example, in a gravitational field, if you roll downhill, your potential energy decreases. Second, and more seriously, it is easy to see that this specific force field, as it is formulated on a lattice, is not conservative. For example, consider a system where one particle is fixed at the origin, and a second particle undergoes a cyclic motion from

$(1,0) \rightarrow (2,0) \rightarrow (2,1) \rightarrow (1,1) \rightarrow (1,0)$. Then the total internal energy change after the cycle is not zero, even though the final and initial configuration are exactly the same. Therefore, this lattice force field cannot be derived from a potential.

Entropy: The probability of moving to a neighbor lattice site is associated with an entropy change, which is estimated by the sum of step lengths to unoccupied neighbor cells. This specific form of entropy is entirely heuristic and again, it cannot be derived from an effective entropy potential. One should also note that it is not necessary to include translational entropy in a proper

Monte Carlo algorithm: The algorithm will automatically account for it.

Jump probability (page 8): From the previous two points, it is already clear that the quantity $\Delta A$ in the expression for p cannot be associated with a well-defined effective potential A. However, even if such a potential existed, the choice of jump probabilities seems rather arbitrary. For example, page 8 says "If $0 < p <$

1, the destination cell is chosen at random from all those with the same largest jump probability $p < 1$". This is not well motivated. Why not choose from all cells with weighted probabilities according to their jump probability? The algorithm described here is not motivated by any microscopic considerations.

With the same right, assuming that $\Delta A$ could really be derived from a global effective potential function $A$, one could also use a standard Metropolis algorithm, which would satisfy detailed balance by construction.

Reply: Thank you for pointing out the sign issue with the internal energy. It has been corrected and is explained in the supplement: "The internal energy change $\Delta U_{f,i} =$

$-(F_f - F_i)\Delta R_{f,i} \approx F_i \Delta R_{f,i}$ is modeled for each potential jump from the initial occupied cell $i$ to the final empty cell $f$ by the product of the net force $F_i$ with the vector $\Delta R_{f,i}$

connecting the centers of the initial cell $i$ and the final cell $f$."

Thank you also for pointing out that the force field underlying the definition of the internal energy is not conservative. This clarifies that energy is imparted or extracted from the system at every jump, so that the jumps are not in thermal equilibrium but rather in a driven equilibrium. The following sentence has been added (lines 206-210):

"It is noted here that the force field on a randomly populated lattice is not conservative (Reviewer, 2023). In other words, the energy balance of a particle moving in a circle is different from zero, and Monte Carlo simulations under these constraints probe a driven equilibrium and not thermodynamic equilibrium (Michel et al., 2014)."

Entropy: Your remarks are appreciated. The heuristic nature of our entropy term has been addressed in the reply to the comments of reviewer #1.

Jump probability: With reference to our last reply, it is added that the same jump
probability $p < 1$ can be obtained for jumps to different cells. When this is the case, the
destination cell is chosen at random from this subset. See supplement, lines 56–57.

3c) Summary: The presented Monte Carlo algorithm does not satisfy detailed
balance for two reasons: First, even though the notation suggests otherwise,
the underlying quantities $p$ are not associated with a well-defined effective
energy function $A$. Second, the jump probabilities are chosen heuristically
according to some random rules which are not well-motivated. It is not
surprising that these rules do not satisfy detailed balance, because imposing
detailed balance usually requires special efforts.

In fact, these rules would not even guarantee global balance if $A$ were well-
defined. On the other hand, they do define some kind of stochastic Markovian
dynamics, and according to the central limit theorem of finite Markov systems,
the probability distribution will converge against some stationary fixed point,
which however differs from the Boltzmann distribution $N \exp(-\beta A)$.
Furthermore, this stationary state would include persistent currents by default,
because, as explained above, special efforts must be taken to remove them in
such a model.

Reply: We agree that the model does not apply to thermodynamic equilibrium and thus
to detailed balance. Since energy is not conserved when introducing a jump probability,
but mass balance is obeyed, the model applies to driven and not thermodynamic
equilibrium. We believe that this is still an interesting conclusion, as it suggests, that
molecular motion in pores can be driven into circular exchange by external forces
imparted by electric, magnetic or mechanical (ultrasonic) fields either broadband at
multiple frequencies or narrow band at a single frequency. If proven experimentally,
chemical reactions accelerated by heterogeneous catalysts could be improved.

4) Interpretation of the results:

4a) Thermodynamic equilibrium: The term "thermodynamic equilibrium", by
definition, refers to a stationary state without currents. One of the central
postulates of thermodynamics is that every physical closed dynamical system
reaches thermodynamic equilibrium at some point. This is a postulate and might be debated. However, a system with persistent currents as described in
the manuscript would not be considered to be at thermodynamic equilibrium.

Reply: We understand.

4b) Detailed balance and nonequilibrium thermodynamics: As correctly stated in
the manuscript, the lack of currents is associated with microscopic detailed
balance – or, putting it the other way round, breaking detailed balance normally
generates currents. However, this also implies that entropy is constantly being
produced, and dissipated, see, e.g., References [1-3].

Reply. We understand. In view of this issue, we refer to Feynman's rachet in the
manuscript at line 160. Thank you also for the literature references! They are cited in
the revised manuscript.

4c) Dynamical systems with broken detailed balance have been discussed in
nonequilibrium thermodynamics for many decades. Physically, they are used
to describe open dissipative systems, for example, living systems or active
systems [1,2], which are stabilized via a steady input of energy. It is easily
possible to design stochastic dynamical systems that break detailed balance,
as has been done, e.g., in the present manuscript or in Refs. [3,4]. In Monte
Carlo simulations, implementing such dynamics can have the advantage that
a desired probability distribution function can be sampled much more efficiently
[4].

Reply: Thank you for clarifying. We fully agree. The manuscript has been revised
accordingly.

4d) Detailed balance and Monte Carlo: The Monte Carlo method has been
introduced by Metropolis et al as a method to efficiently sample a desired target
probability distribution. The necessary ingredient for this is to impose global
balance. Detailed balance is not strictly necessary. With the exception of
kinetic Monte Carlo (which has not been used here), Monte Carlo dynamics is
typically not realistic. Nevertheless, Monte Carlo is also used to study
dynamical systems in a coarse-grained sense. However, it is important to note
that in this type of model, you get out what you put in. If you implement Monte

Carlo moves that break detailed balance, then clearly, you will find that detailed
balance is broken in your system. Therefore, Monte Carlo simulations
designed to model dynamics at thermal equilibrium must be set up such that
the Monte Carlo moves satisfy detailed balance.

Reply: Agreed. Thank you. Reference to Metropolis et al. is now made at several
occasions in the revised manuscript.

4e) Is detailed balance always fulfilled? As stated above, the claim that closed
physical dynamical systems reach thermodynamic equilibrium is a postulate.
It lies at the heart of the second law of thermodynamics, but being a postulate,
it could be violated in certain cases. In fact, it is violated, e.g., for integrable
systems such as linear harmonic chains. It has not been proved rigorously
except for a few special cases. On the other hand, the opposite claim that
detailed balance might be broken in realistic (closed) physical system
fundamentally challenges the foundations of thermodynamics. Such a claim
cannot be based on Monte Carlo simulations. This is because, as explained
above, Monte Carlo dynamics are inherently artificial, and it is much easier to
implement dynamical models that break detailed balance than to implement
models that satisfy detailed balance. The claim would have to be based on
experiments, or on molecular simulations of a truly microscopic model, e.g.,
classical Hamiltonian dynamics or Schrödinger dynamics. In fact, there have
been several claims in the past, based on atomistic simulations, that the
second law might be broken in nanoscale systems. For example, spontaneous
unidirectional currents through pores or the like were observed in simulations.
In all of these cases, it eventually turned out that the claimed effects could be
attributed to numerical artefacts of the simulations.

Reply: Thank you for these explanations!

4f) The central question is whether a system can thermalize, which is a valid
question especially for nanoscale systems and subject of active research.
Specifically, the gas diffusion case discussed in the manuscript is related to
the question whether a classical ideal gas can thermalize. This is one of the
few cases which has been studied very intensely and for which rigorous results
exist (the H-theorem, see [5]). Ideal gases do thermalize! In the manuscript, nonideal gases with excluded volume interactions are considered, which might change the situation, but I would be very surprised if it did. This is one of the reasons why it is so important that the authors describe their simulations for the gas diffusion simulations in more detail. If they maintain the claim that detailed balance is broken in these (off- lattice) systems, they should prove it much more carefully, e.g., by systematic variation of the time step, by studying the relaxation of several quantities as a function of simulation time, and by a solid assessment of error bars.

References:
1 C. W. Lynn et al, PNAS 2021, 118, e2108998118.
2 F. S. Gnesotto et al, Rep. Prog. Phys. 2018, 81, 066601.
3 L. Crochik et al, Phys. Rev. E 2005, 72, 057103.
4 M. Michel et al, J. Chem. Phys. 2014, 140, 054116.
5 G. Truesdell, R.G. Muncaster, Fundamentals of Maxwell's kinetic theory of
a simple monatomic gas, Chapter XI, in Pure and Applied Mathematics,
Volume 83, pp. 145-172 (1980).

Reply. The thermalization of the of the gas-diffusion algorithm had been tested but not mentioned in the original manuscript. The algorithm is now described in more detail in lines 220-248. Moreover, the asymmetry parameter as a function of the position of the active site in the wall of the small square pore has been evaluated at two different time steps (Fig. 6e). It is found that the asymmetry parameter decreases significantly with decreasing time step, indicating that it approaches zero for infinitesimally small time.

Moreover, it is found, that the uneven distribution of average density inside the pore obtained with the gas-diffusion algorithm results from projecting the particle positions at the time of observation onto a course grid and not at the exact collision time.

5)  More technical issues.

5a) I still think there has to be an equation for the asymmetry parameter which can be understood by everybody. Do I understand correctly that you average the quantity given in (5) over all jumps from 2->3 during the simulation, but you give different weights depending on the initial (or final, or both?) position of the particle (whether it is close to an active site or not?)

Reply: There are no different weights assigned to particles in different pools. Please see our response in 2a).

5b) Thanks for clarifying the details of the off-lattice simulation. It is of course ok and common practice to use simulation units and not SI units. Your simulation units are apparently defined in terms of the mass m of the particles, the particle diameter σ (I assume it is one in your units), and the energy (I assume you set

$kT$=1 when setting up the Maxwell-Boltzmann velocity distribution). This defines the time unit $\tau = (m\, s^2\, /kT)^{1/2}$ . In these units, your time step is $\Delta t = 1$ τ, which is very large. In molecular dynamics simulations, typical values are $\Delta t =$

$10^{-3}$ τ or less, and having a too large time step can have a severe impact on the results.

Reply: We agree that the time step has been too large and as a result has introduced a false image of asymmetry by an overlap of the hard circles and by possibly skipping collisions. The simulation has now subsequently been tested at shorter time steps to determine if such an error was introduced. We find that the asymmetry parameter gets smaller as the time step is decreased. The result is reported in Fig. 6e.

5c) On the other hand, you do not really describe a Molecular Dynamics simulation, it is rather another type of (off-lattice) Monte Carlo simulation. For example, your collisions preserve the energy of the two colliding particles, but not their momentum. As a Monte-Carlo simulation, it does not preserve detailed balance, and therefore, again, it is not surprising that the results also break detailed balance.

Reply: The gas phase simulation (off-lattice simulation) is a common time-driven elastic hard circle model with walls rather than periodic boundaries. An initial distribution of speeds is generated, and the particles are given a random initial direction of travel. After a collision, new velocities and deflection angles for the two particles are determined from conservation of momentum and kinetic energy as described in the response 2c). As mentioned in the previous response, the asymmetry parameter decreases with shorter time step suggesting that detailed balance is recovering at shorter observation intervals. Including event driven dynamics into this code should help in the study of detailed balance violation.

5d) Regarding the comment "We found nonzero asymmetry parameters also when choosing the destination cell for a jump from all vacant neighbor cells at random." I would like to note that this algorithm also breaks detailed balance.

In order to maintain detailed balance, you have to choose a jump randomly from **all** neighbor cells and then reject the move if the neighbor cell is filled.

Rejecting means the particle stays where it is and does not move at all, but the follow-up configuration still counts for the overall statistics. But I suppose the authors are aware of this, since they also state " Choosing an algorithm that satisfies detailed balance by construction precludes testing detailed balance"

(a statement to which I fully agree.)

Reply: Thank you for clarifying! We reran the calculations for Fig. 6e choosing randomly from all neighbor positions, free and occupied. Indeed, the asymmetry parameter produces noise more than one order of magnitude lower than the values observed with zero probability assigned to randomly chosen jumps to occupied positions. We now understand, that by introducing a probability to jumps, detailed balance is violated and cite the Metropolis reference. So, we are driving the imbalance by our vacancy diffusion algorithm. Based on this understanding the entire manuscript underwent major revision.

5e) Regarding the comment on currents: Cyclic exchange, as long as it is persistent and does not average to zero on the long run, would also count as current in the definition of thermal equilibrium.

Reply: ok.

5f) Regarding thermalization: This not only means thermalization with respect to velocities. Asking whether a system thermalizes is the same as asking whether a system reaches thermal equilibrium in the above sense, i.e., a stationary state without stationary (macroscopic or microscopic) currents. Testing this is generally difficult in a simulation. The velocity distribution usually approaches the Maxwell-distribution very quickly, but other quantities usually equilibrate much more slowly. In your simulations, you could for example test the system without force terms and check whether the system ever reaches the Boltzmann distribution with respect to positions. In the absence of any forces, the particles should be uniformly distributed in the pore. This is probably not the case.

Reply: Thermalization of the speed distribution has been verified for the gas-phase (off-lattice) simulations which is devoid of force terms. But we understand now that we are observing a driven equilibrium because the algorithm cannot exactly catch the
instant of a particle collision. We are also observing a driven equilibrium with the
vacancy-diffusion lgorithm in the absence of force terms but the presence of a jump
probability. In both cases, the population density across the pore shows oscillations.
Experimental evidence (Song 2000) and tested theory (Brownstein 1977) indicate the
existence of diffusion eigenmodes of fluids confined to pores, which describe spatially
oscillating distributions of nuclear magnetization components in when excited away
from thermodynamic equilibrium. These decay in distinct ways under the impact of
diffusion and the boundary geometry. Considering, that three-site exchange probes
Fick's second law, we interpret our observed currents, to be a pore-resonance effect
on translational motion which relates to diffusion eigenmodes.

**Conclusion**

With the lessons learned in the open discussion of our submission and in consideration
of the expert advice of the reviewers we have substantially revised our manuscript.
The first point is that the reported computer simulations do not indicate a violation of
the principle of detailed balance so that there is no indication for three-site NMR
exchange maps to be asymmetric in thermodynamic equilibrium. If observed anyway
the asymmetry needs to be attributed to experimental deficiencies or artifacts from data
processing (e. g. inverse Laplace transformation). The second point is that the
observed particle dynamics obey the diffusion equation and appear to be linked to a
diffusion eigenmode with the consequence that diffusion eigenmodes may possibly be
driven by external stimuli like the violin bow enforcing resonance vibration of a Chladni
plate. This is a technically interesting perspective as heterogeneous catalysis may
possibly be enhanced be oscillating electrical, magnetic, or mechanical (ultrasonic)
fields.